# Early postnatal serotonin modulation prevents adult-stage deficits in *Arid1b*-deficient mice through synaptic transcriptional reprogramming

Hyosang Kim [1,8], Doyoun Kim [2,8], Yisul Cho [3,8], Kyungdeok Kim [2], Junyeop Daniel Roh [2], Yangsik Kim [4], Esther Yang [5], Seong Soon Kim [6], Sunjoo Ahn [6], Hyun Kim [5], Hyojin Kang [7], Yongchul Bae [3] ✉ & Eunjoon Kim [1,2] ✉

Autism spectrum disorder is characterized by early postnatal symptoms, although little is known about the mechanistic deviations that produce them and whether correcting them has long-lasting preventive effects on adult-stage deficits. ARID1B, a chromatin remodeler implicated in neurodevelopmental disorders, including autism spectrum disorder, exhibits strong embryonic- and early postnatal-stage expression. We report here that *Arid1b*-happloinsufficient (*Arid1b^{+/−}*) mice display autistic-like behaviors at juvenile and adult stages accompanied by persistent decreases in excitatory synaptic density and transmission. Chronic treatment of *Arid1b^{+/−}* mice with fluoxetine, a selective serotonin-reuptake inhibitor, during the first three postnatal weeks prevents synaptic and behavioral deficits in adults. Mechanistically, these rescues accompany transcriptomic changes, including upregulation of FMRP targets and normalization of HDAC4/MEF2A-related transcriptional regulation of the synaptic proteins, SynGAP1 and Arc. These results suggest that chronic modulation of serotonergic receptors during critical early postnatal periods prevents synaptic and behavioral deficits in adult *Arid1b^{+/−}* mice through transcriptional reprogramming.

Neurodevelopmental disorders, including autism spectrum disorders (ASD) and intellectual disability (ID), are characterized by early postnatal manifestation of symptoms. Core symptoms of ASD include social deficits and repetitive behaviors, which accompanies various comorbidities, including intellectual disability, hyperactivity, anxiety, epilepsy, and sensory abnormalities. Previous human genetic studies have identified a large number of ASD-risk genetic variations with diverse neural functions[1–5], although major challenges still remain in areas, including correlating symptomatic heterogeneity with genetic variations[6] and deepening our understanding of ASD pathophysiology for the development of efficient treatments[2,7,8].

[1]Department of Biological Sciences, Korea Advanced Institute for Science and Technology (KAIST), Daejeon 34141, Korea. [2]Center for Synaptic Brain Dysfunctions, Institute for Basic Science (IBS), Daejeon 34141, Korea. [3]Department of Anatomy and Neurobiology, School of Dentistry, Kyungpook National University, Daegu 41940, Korea. [4]Graduate School of Biomedical Engineering, Korea Advanced Institute for Science and Technology (KAIST), Daejeon 34141, Korea. [5]Department of Anatomy and Division of Brain Korea 21, Biomedical Science, College of Medicine, Korea University, Seoul 02841, Korea. [6]Therapeutics and Biotechnology Division, Korea Research Institute of Chemical Technology (KRICT), Daejeon 34114, Korea. [7]Division of National Supercomputing, Korea Institute of Science and Technology Information, Daejeon 34141, Korea. [8]These authors contributed equally: Hyosang Kim, Doyoun Kim, Yisul Cho. ✉ e-mail: ycbae@knu.ac.kr; kime@kaist.ac.kr

Many ASD-risk genes encode proteins that regulate early brain development such as chromatin remodelers and transcriptional factors[1–5], that are strongly expressed at embryonic and early postnatal stages. This suggests that ASD-related mechanistic deviations likely occur at early developmental stages. Importantly, these early postnatal stages coincide with a critical period of brain development during which the brain is flexible and accessible to external/epigenetic modulation[9,10]. Therefore, these ASD-related mechanistic deviations could be identified early and corrected to minimize secondary deviations that gradually precipitate stronger adult-stage deficits[11].

ARID1B (AT-rich interaction domain 1B; also known as BAF250B) is a member of the SWI/SNF family of chromatin remodelers. Haploinsufficiency of ARID1B, involving various genetic variations such as missense/frame-shift/nonsense point mutations and protein-truncating intragenic deletions, has been frequently associated with ASD, ID, and Coffin-Siris Syndrome, a genetic disorder accompanying arrays of developmental and cognitive delays[12–22]. Previous studies on Arid1b-deficient mice reported developmental and behavioral deficits relevant to ARID1B-related diseases and the mechanisms that likely underlie the defective phenotypes in mice, including suppressed inhibitory synaptic transmission and insulin-like growth factor signaling[23–26]. However, because mutations in the same gene can induce distinct pathological mechanisms depending on the nature of the mutation, genetic background of the animal species used (i.e., mouse strain), model system employed (mouse vs. human neurons), and developmental stage examined (i.e., early vs. late postnatal), our search for the mechanisms underlying phenotypes of Arid1b-deficient mice may be at an early stage.

In the present study, we found persistently decreased excitatory, but not inhibitory, synaptic deficits in mice with heterozygous deletion of the Arid1b gene (exon 5; Arid1b$^{+/-}$ mice), which display social and repetitive behavioral deficits. Early postnatal, chronic fluoxetine treatment of Arid1b$^{+/-}$ mice prevented the appearance of excitatory synaptic and behavioral deficits in adult Arid1b$^{+/-}$ mice through mechanisms that include upregulation of FMRP (fragile X mental retardation protein) targets and normalization of HDAC4/MEF2A-related transcriptional regulation of the synaptic proteins, SynGAP1 and Arc. These results suggest that serotonergic receptor modulation during a critical early postnatal period prevents adult-stage deficits through transcriptomic reprogramming.

## Results

### Autistic-like social and repetitive behavioral deficits in Arid1b$^{+/-}$ mice

To explore the mechanisms underlying ARID1B-related neurodevelopmental and psychiatric disorders, we generated Arid1b-haploinsufficient mice carrying a protein-truncating mutation at the start of exon 5 encoding the atrophin-1 domain of the protein and analyzed the resulting Arid1b$^{+/-}$ mice using transcriptomic, electrophysiological, and behavioral approaches (Fig. 1a–c; Supplementary Fig. 1a–c). These mice were born in largely normal Mendelian ratios, but showed moderately decreased body weights from postnatal day 9 (P9) compared with wild-type (WT) littermates (Supplementary Fig. 1d; Supplementary Table 1). The gross morphology of the brain was largely normal, as determined by the assessment of sub-brain areas and cortical thicknesses (total and superficial/deep layers) (Supplementary Fig. 2), although there were moderate increases in the areas of lateral (but not third) ventricles and moderate decreases in corpus callosal areas and apical (not basal) dendritic complexity in hippocampal neurons, measured at P42 using Arid1b$^{+/-}$; Thy1-EGFP mice (Supplementary Fig. 3).

In behavioral tests, Arid1b$^{+/-}$ mice showed decreased direct social interaction in the dyadic social-interaction test but normal social approach and social novelty recognition in the three-chamber social-interaction test (Fig. 1d, e). As pups (P3–11), these mice also showed decreased ultrasonic vocalizations (USVs) when separated from their mothers (Fig. 1f), indicative of impaired social communication. Additionally, they displayed enhanced repetitive self-grooming (Fig. 1g) but normal levels of locomotor activity (open-field test), anxiety-like behavior (elevated plus-maze and light-dark tests), and recognition memory (novel object-recognition test) (Supplementary Fig. 4). Therefore, Arid1b$^{+/-}$ mice show strong autistic-like social and repetitive behavioral deficits.

### Synapse- and ASD-related transcriptomic changes in the whole brain and mPFC of Arid1b$^{+/-}$ mice

To explore molecular and cellular mechanisms underlying the behavioral deficits in Arid1b$^{+/-}$ mice at pup and adult stages, we first characterized the transcriptomic changes at P10, a postnatal stage in which Arid1b expression decreases sharply (Supplementary Fig. 5a, b). In addition to the whole brain, we included the medial prefrontal cortex (mPFC), a brain region strongly implicated in ASD[27]. We also used whole-brain samples at P3, an earlier, postnatal time point. RNA-Seq analyses of these samples from Arid1b$^{+/-}$ and WT mice (Supplementary Dataset 1) were performed using gene set enrichment analysis (GSEA), which uses the whole list of genes with altered transcriptional levels to minimize the bias that can result from analysis of a small fraction of differentially expressed genes (DEGs) above an artificial cutoff (p-values or fold change)[28]. The numbers of DEGs with significant FDR values (<0.05) in the P3 whole-brain and P10 whole-brain/mPFC transcripts were small (Supplementary Fig. 6a–c), making it difficult to identify significant biological functions.

A comparison of transcriptomes from the whole-brain of Arid1b$^{+/-}$ and WT mice (expressed as the ratio of Arid1b$^{+/-}$ to WT transcripts [Arid1b/WT]) at P10 showed positive enrichment—strong enrichment in the case of upregulated genes—for synapse-related gene sets and negative enrichment for mitochondria- and ribosome-related genes, as reflected in the top-five enriched gene sets in Cellular Component (CC) and Biological Process (BP) domains of the Gene Ontology (GO) database (Fig. 2a; Supplementary Dataset 2). Integration and visualization of the whole (not top-five) GSEA results using the Cytoscape EnrichmentMap App[29] further highlighted these biological functions (Fig. 2a).

In the whole brain at P3, the comparison of Arid1b$^{+/-}$ and WT transcripts indicated enrichment patterns that are similar to those observed in the whole brain at P10, including (1) positive enrichment for synapse- and ribosome/mitochondria-related gene sets, and (2) negative enrichment for chromatin-related gene sets, as shown in the top-five enriched gene sets and Cytoscape EnrichmentMap results (Supplementary Fig. 7a; Supplementary Dataset 2).

In the mPFC at P10, the comparison of Arid1b$^{+/-}$ and WT transcripts indicated patterns that are distinct from those observed in the whole brain at P10, including (1) positive enrichment for ribosome- and extracellular matrix-related gene sets, (2) negative enrichment for synapse-related gene sets, and (3) positive enrichment for inflammation/immune-related gene sets (Fig. 2b; Supplementary Dataset 2).

When tested for ASD-related/risk gene sets, Arid1b/WT whole-brain transcripts at P10 were enriched towards a direction largely opposite that observed in ASD (hereafter termed "reverse-ASD"), showing (1) negative enrichment for ASD-related gene sets that are upregulated in ASD (DEG_Up and Co-Exp_Up)[30], (2) positive enrichment for gene sets that are downregulated in ASD (DEG_Down)[30], and (3) positive enrichment for ASD-risk gene sets that are thought to be downregulated in ASD through loss-of-function mutations such as SFARI (all or high-confidence), FMRP targets, DeNovoMis (protein-disrupting or missense rare de novo variants/RDNVs), DeNovoVariants (protein-disrupting rare de novo variants/RDNVs), and AutismKB[31–35] (Fig. 2c).

In addition, for cell-type–specific gene sets that are distinctly associated with ASD (decreased neuronal and oligodendrocytic gene expression and increased astrocytic/microglial gene expression)[31,36–40], Arid1b/WT whole-brain transcripts were positively enriched for neuron

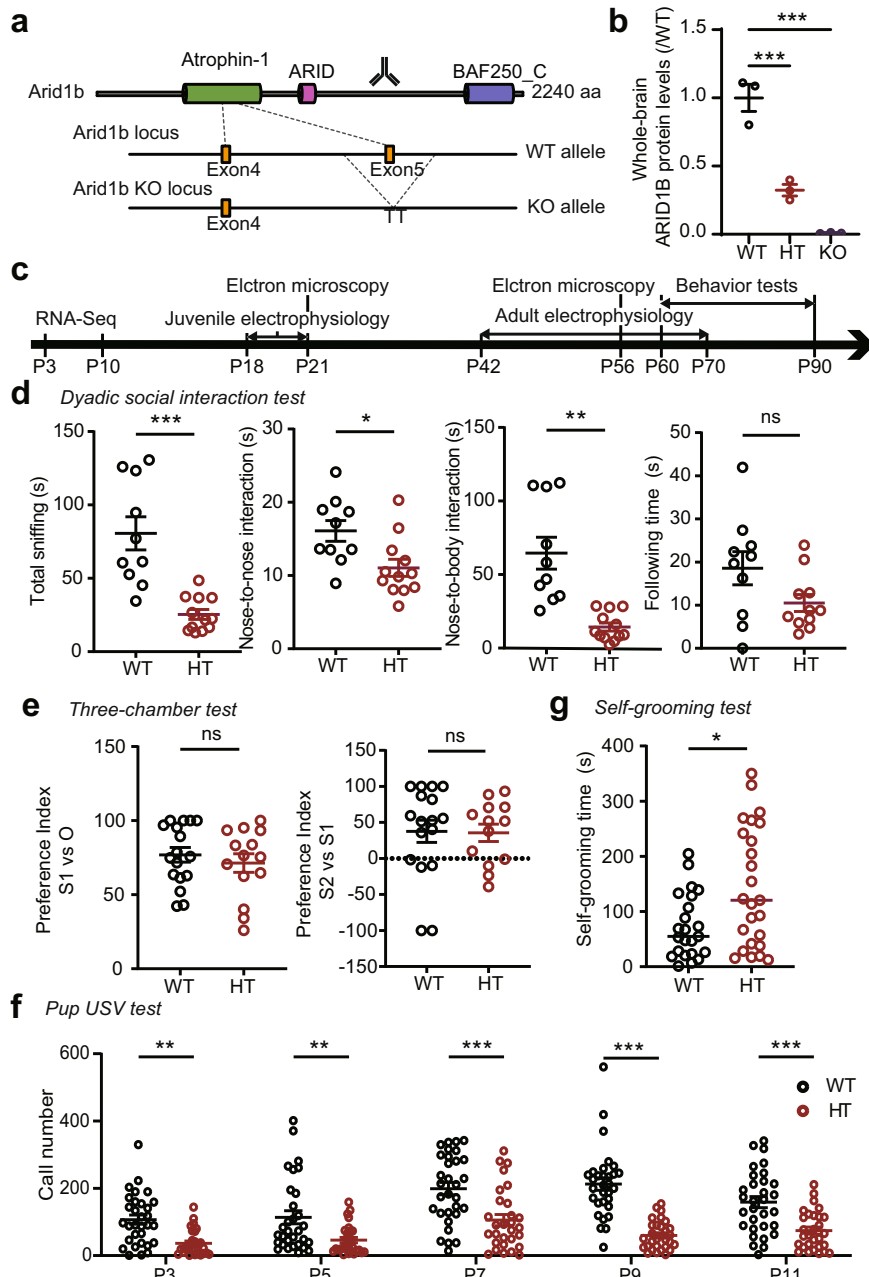

**Fig. 1 | Autistic-like social and repetitive behavioral deficits in *Arid1b*^+/− mice.**
**a** Strategy for *Arid1b* knockout (KO) in mice. A protein-truncating mutation was introduced by inserting "TT" nucleotides at the starting region of exon 5, which encodes part of the atrophin-1 domain. The antibody-targeting region is indicated. Atrophin-1, ARID, and BAF250_C indicate specific domains of the Arid1b protein. **b** Reduced levels of ARID1B protein in *Arid1b*^+/− (HT) and *Arid1b*^−/− (KO) mice, respectively, at P0. ($n$ = 3 mice [WT, HT, and KO], ordinary one-way ANOVA with Dunnett's test, [WT vs HT: $p$ = 0.0005, WT vs KO: $p$ < 0.0001]). **c** Postnatal stages at which RNA-Seq, electrophysiological, electron microscopic, and behavioral experiments were performed in analyses of *Arid1b*^+/− mice. **d** Decreased direct social interaction of *Arid1b*^+/− mice (2–3 months) in the dyadic social-interaction test, as shown by total time spent in social interaction (nose-to-nose sniffing, nose-to-body sniffing, and following). ($n$ = 10 pairs of mice [WT], 12 [HT], two-tailed Student's t-test [total sniffing: $p$ < 0.0001, nose-to-nose: $p$ = 0.0117, following: $p$ = 0.0697],

two-tailed Welch's test [nose-to-body: $p$ = 0.0012]). **e** Normal social approach and social novelty recognition in *Arid1b*^+/− mice (2–3 months) in the three-chamber social-interaction test, as shown by the preference index (time spent sniffing S1 [social target] − O [object] or S2 [novel social target] − S1 [familiar social target] divided by total time spent sniffing). ($n$ = 17 [WT], 13 [HT], two-tailed Student's t-test). **f** Decreased ultrasonic vocalization (USV) calls in *Arid1b*^+/− pups (P3–11) separated from their mothers, measured as the number of calls. ($n$ = 14 mice [male-WT], 17 [male-HT/heterozygote], 17 [female-WT], and 12 [female-HT, two-way ANOVA with Sidak's test [P3: $p$ = 0.0040, P5: $p$ = 0.0062, P7: $p$ < 0.0001, P9: $p$ < 0.0001, P11: $p$ = 0.0003]). **g** Enhanced self-grooming in *Arid1b*^+/− mice (2–3 months) in a novel home cage without bedding. ($n$ = 23 [WT], 25 [HT], two-tailed Mann–Whitney test, $p$ = 0.0322). Graphical data are presented as means ± SEM (*$p$ < 0.05, **$p$ < 0.01, ***$p$ < 0.001, ns, not significant). Source data are provided as a Source Data file.

(both glutamatergic and GABAergic)-related gene sets and negatively enriched for microglia-related gene sets—a strongly reverse-ASD pattern—although oligodendrocyte-related gene sets were negatively enriched, a pattern hereafter termed "ASD-like" (Fig. 2d).

Arid1b/WT whole-brain transcripts at P3 indicated a reverse-ASD pattern that are largely similar to those observed in P10 Arid1b/WT whole-brain transcripts, including (1) negative enrichment for gene sets that are upregulated in ASD (Co-Exp_Up), (2) positive

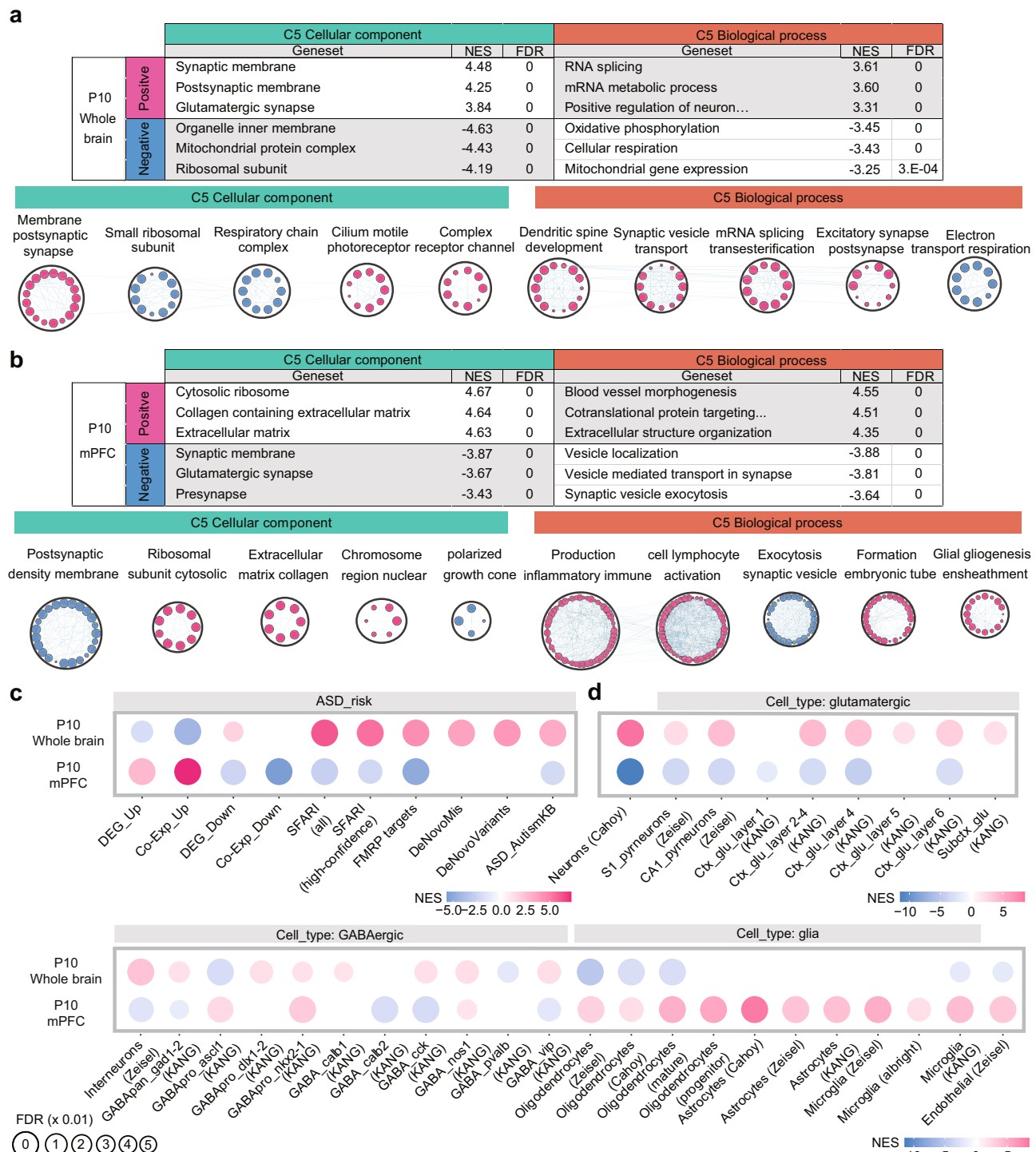

**Fig. 2 | Synapse- and ASD-related transcriptomic changes in the whole brain and mPFC of *Arid1b*[+/−] mice at P10.** List of top-five gene sets (ranked by *p*-value) positively or negatively enriched among Arid1b/WT transcripts from the whole-brain (**a**) and mPFC (**b**) of P10 males, determined by GSEA. CC and BP, cellular component (CC) and biological process (BP) domains in the C5 gene-set database (top), and functional clustering of the enriched gene sets from the GSEA results using Cytoscape EnrichmentMap (bottom). Only the top-five clusters are shown to save space. NES normalized enrichment score; FDR false discovery rate. (*n* = 5 mice [WT and HT]). **c** GSEA of Arid1b/WT P10 whole-brain and mPFC transcripts for ASD-related/risk gene sets. (*n* = 5 mice [WT/HT]). **d** GSEA of Arid1b/WT P10 whole-brain and mPFC transcripts for cell-type-specific gene sets. (*n* = 5 mice [WT/HT]).

enrichment for gene sets that are downregulated in ASD (DEG_Down and Co-Exp_Down), (3) positive enrichment for the FMRP target gene set, and (4) negative enrichment for microglia-related gene sets, although neuron (glutamate and GABA)-related genes were both positively and negatively enriched (Supplementary Fig. 7b, c).

In contrast with these results, Arid1b/WT mPFC transcripts displayed strong ASD-like patterns, showing (1) positive enrichment for gene sets that are upregulated in ASD (DEG_Up and Co-Exp_Up), (2) negative enrichment for gene sets that are downregulated in ASD (DEG_Down and Co-Exp_Down), (3) negative enrichment for ASD-risk

gene sets such as SFARI (all or high-confidence), FMRP targets, and AutismKB, (4) negative enrichment for neuron (glutamatergic)-related gene sets, and (5) positive enrichment for astrocyte/microglia-related gene sets (Fig. 2c, d).

These results collectively suggest that Arid1b/WT whole-brain transcripts at P10 and P3 show similar enrichment patterns with respect to biological functions and ASD-related/risk gene expressions (reverse-ASD), whereas Arid1b/WT mPFC transcripts at P10 show largely opposite transcriptomic patterns.

### Persistently decreased excitatory synaptic density and transmission in prefrontal $Arid1b^{+/-}$ neurons

Changes in synapse-related genes among mPFC Arid1b/WT transcripts suggest that synapse development and function may be altered in $Arid1b^{+/-}$ mice. To test this, we measured excitatory and inhibitory synaptic transmission in the medial prefrontal cortex (mPFC).

Layer 2/3 pyramidal neurons in the prelimbic region of the mPFC in juvenile (~P21) $Arid1b^{+/-}$ mice showed markedly decreased frequency (~50%), but not amplitude, of miniature excitatory postsynaptic currents (mEPSCs) (Fig. 3a). In contrast, miniature inhibitory postsynaptic currents (mIPSCs) were not changed (Fig. 3b). Electron microscopic analysis indicated a decrease (~30%) in the density, but not morphology (length, thickness, and perforation), of the postsynaptic density (PSD), representing excitatory postsynaptic structures (Fig. 3c–g). These results suggest that the decrease in mEPSC frequency mainly involves decreased excitatory synapse density, in line with the decreased synapse-related gene expression in the mPFC at ~P10.

These changes in mEPSCs persisted into adulthood, where the frequency (but not amplitude) of mEPSC was decreased in layer 2/3 pyramidal neurons (~P42), while mIPSCs were unaltered (~P56) (Fig. 3h, i). Paired-pulse facilitation was normal at these synapses (Fig. 3j), suggesting normal presynaptic release. Intrinsic excitability was also normal, as shown by current-firing curve, action potential threshold, input resistance, and resting membrane potential (Fig. 3k–n). Layer 5 pyramidal neurons from $Arid1b^{+/-}$ mice showed no alterations in mEPSCs or mIPSCs (Supplementary Fig. 8a, b) indicative of cortical layer-specific changes. Similarly, layer 6 $Arid1b^{+/-}$ pyramidal neurons showed unaltered mEPSCs (Supplementary Fig. 8c). In the mutant anterior cingulate cortex (ACC), layer 2/3 or layer 5 pyramidal neurons did not show alterations in mEPSCs (Supplementary Fig. 8d, e).

To further investigate the lack of changes in mIPSCs in layer 2/3 pyramidal neurons in the mutant mPFC, we measured spontaneous inhibitory postsynaptic currents (sIPSCs), which represent inhibitory synaptic transmissions under neuronal network activity, but could not detect a genotype difference between WT and $Arid1b^{+/-}$ mice (~P56) (Supplementary Fig. 8f). In addition, electron microscopic anlyses indicated a normal density of inhibitory synapses in the mPFC (layer 2/3) of $Arid1b^{+/-}$ mice (~P56) (Supplementary Fig. 8g).

Given that persistent excitatory synaptic deficits were observed in both juvenile and adult prefrontal pyramidal neurons, we tested these excitatory neurons for the expression of $Arid1b$ mRNA using fluorescence in situ hybridization (FISH). $Arid1b$ mRNA was detected in both glutamatergic and GABAergic neurons in cortical areas, including the retrosplenial cortex, at P7, P14, and P56 (Supplementary Fig. 9). However, there was no change in the number of parvalbumin-positive GABA neurons (Supplementary Fig. 8h), a major GABA neuron subtype in the cortex[41].

These results collectively suggest that a haploinsufficiency of $Arid1b$ selectively and persistently decreases excitatory, but not inhibitory, synaptic density and transmission in the mPFC at both juvenile and adult stages.

### Early, chronic fluoxetine treatment prevents behavioral deficits in adult $Arid1b^{+/-}$ mice

Increasing evidence from mouse models of ASD indicates that early mechanistic deviations lead to ASD-like phenotypes in adults, and that early correction of these deficits produces long-lasting effects[11]. We hypothesized that the decreased excitatory synaptic density and transmission that was first observed at juvenile stages in $Arid1b^{+/-}$ mice may underlie the autistic-like behavioral deficits in adults. To test this hypothesis, we attempted an early-stage pharmacological rescue and assessed its long-lasting effects in $Arid1b^{+/-}$ mice.

To this end, we employed chronic treatment with fluoxetine, which has been shown to enhance excitatory synaptic transmission in rodents[42,43] and ameliorate ASD symptoms in humans[44] and ASD-like phenotypes in some mouse models of ASD[45]. $Arid1b^{+/-}$ mice were treated with fluoxetine chronically throughout pup and juvenile stages (P3–21) and their behaviors were tested at adult stages (P60–90) (Fig. 4a). Fluoxetine was delivered to $Arid1b^{+/-}$ pups and juveniles at pre-weaning stages through mother's milk, as previously described[45].

Early fluoxetine treatment rescued social interaction in $Arid1b^{+/-}$ mice, as measured by direct social-interaction scores, without affecting WT mice (Fig. 4b). This treatment also rescued self-grooming in adult $Arid1b^{+/-}$ mice, again, without affecting WT mice (Fig. 4c). Early fluoxetine treatment had no effect on anxiety-like behavior in $Arid1b^{+/-}$ or WT mice, although it moderately decreased body weight and locomotor activity (Supplementary Fig. 10). Fluoxetine treatment in adult $Arid1b^{+/-}$ mice had no effect on social interaction or self-grooming, although it moderately decreased locomotor activity (Supplementary Fig. 11). These results suggest that early, chronic fluoxetine treatment exerts long-lasting preventive effects on social and self-grooming deficits in adult $Arid1b^{+/-}$ mice.

### Early, chronic fluoxetine prevents excitatory synaptic deficits appearing in adult $Arid1b^{+/-}$ mice

Next, to determine whether early fluoxetine-dependent behavioral rescue in adult $Arid1b^{+/-}$ mice involves normalization of excitatory synaptic density and transmission, we performed electrophysiological and electron microscopic experiments.

We found that early, chronic fluoxetine treatment completely rescued the decreased frequency of mEPSCs in $Arid1b^{+/-}$ mice compared with vehicle-treated $Arid1b^{+/-}$ mice (Fig. 5a, b). Early fluoxetine treatment of WT mice did not affect mEPSCs. In addition, early fluoxetine treatment did not affect paired-pulse facilitation in $Arid1b^{+/-}$ or WT mice (Fig. 5c), suggesting that the rescue of mEPSC frequency does not involve normalization of presynaptic release.

Importantly, electron microscopic analyses of excitatory synapse density in vehicle- or fluoxetine-treated WT and $Arid1b^{+/-}$ mice indicated that early fluoxetine treatment rescued the density of the PSD in $Arid1b^{+/-}$ mice without affecting PSD density in WT mice (Fig. 5d–h); it also had no effect on the morphology (length, thickness, and perforation) of the PSD in WT or $Arid1b^{+/-}$ mice.

These results collectively suggest that early, chronic fluoxetine treatment completely rescues excitatory synapse density and function in $Arid1b^{+/-}$ mice without affecting presynaptic release.

### Early fluoxetine prevents aberrant transcriptomic changes in adult $Arid1b^{+/-}$ mice

We next tested if early, chronic fluoxetine treatment prevents aberrant transcriptomic changes in adult $Arid1b^{+/-}$ mice. To this end, we performed RNA-Seq analyses of whole-brain and mPFC samples from vehicle- or fluoxetine-treated (P3–21) WT and $Arid1b^{+/-}$ mice at P120 (Supplementary Dataset 3).

Whole-brain transcripts from vehicle-treated $Arid1b^{+/-}$ and WT mice at P120 (Veh-Arid1b/WT transcripts), representing the baseline genotype difference in adults, were positively enriched for ribosome/mitochondria-related gene sets, as further supported by EnrichmentMap

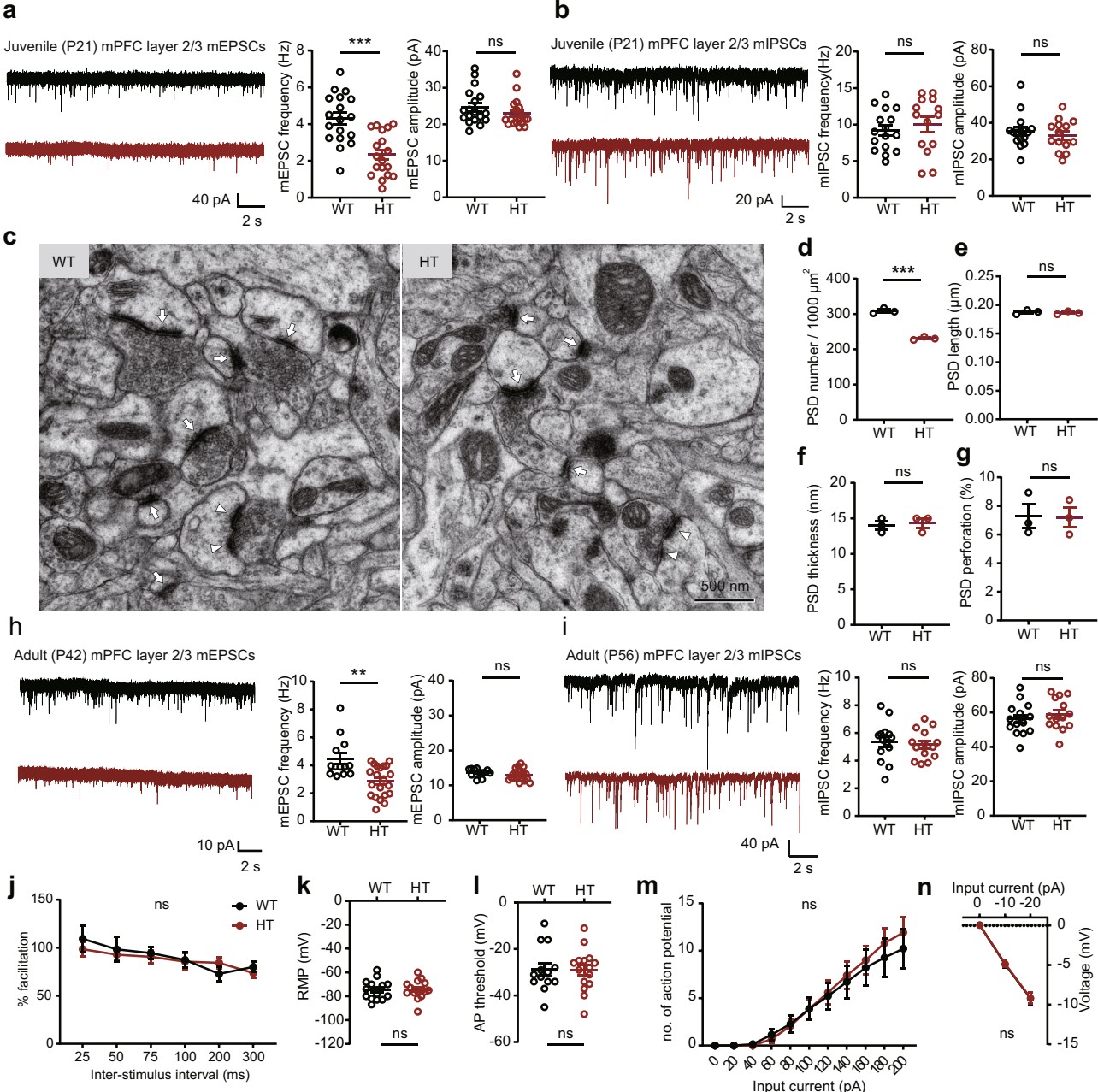

**Fig. 3 | Persistently decreased excitatory synaptic density and transmission in prefrontal *Arid1b*[+/−] neurons. a** Decreased frequency, but not amplitude, of miniature excitatory postsynaptic currents (mEPSCs) in layer 2/3 pyramidal neurons in the prelimbic region of the medial prefrontal cortex (mPFC) of juvenile (P19–21) *Arid1b*[+/−] mice. (n = 18 neurons from 3 mice [WT] and 18, 3 [HT], two-tailed Student's t-test [frequency], two-tailed Mann–Whitney test [amplitude] [frequency: p < 0.0001, amplitude: p = 0.1916]). **b** Normal mIPSC (miniature inhibitory postsynaptic current) frequency and amplitude in layer 2/3 pyramidal neurons in the prelimbic region of the mPFC of juvenile (P19–21) *Arid1b*[+/−] mice. (n = 16, 3 [WT] and 14, 3 [HT], two-tailed Student's t-test). **c–g** Decreased density but normal morphology of postsynaptic densities (PSDs), representing excitatory synaptic structures, in the prelimbic layer 2/3 region of the mPFC of juvenile (P21) *Arid1b*[+/−] mice, as shown by PSD density, length, thickness, and perforation (a measure of excitatory synaptic maturation). (n = 3 mice [WT], 3 [HT], two-tailed Student's t-test [density, length, perforation], two-tailed Mann–Whitney test [thickness], [density: p < 0.0001, length: p = 0.6213, thickness: p = 0.8137, perforation: p = 0.9337]).

**h** Decreased frequency, but not amplitude, of mEPSCs in layer 2/3 pyramidal neurons in the prelimbic region of the mPFC of adult (P42) *Arid1b*[+/−] mice. (n = 13, 3 [WT], 21, 4 [HT], two-tailed Mann–Whitney test, [frequency: p = 0.0053, amplitude: p = 0.2308]). **i** Normal mIPSCs in layer 2/3 pyramidal neurons in the prelimbic region of the mPFC of adult (P56) *Arid1b*[+/−] mPFC mice. (n = 14, 3 [WT], 14, 3 [HT], two-tailed Student's t-test). **j** Normal paired-pulse facilitation in layer 2/3 pyramidal neurons in the prelimbic region of the mPFC of adult (P56) *Arid1b*[+/−] mice. (n = 8, 3 [WT], 6, 3 [HT], two-way ANOVA). **k–n** Normal excitability in layer 2/3 pyramidal neurons in the prelimbic region of the mPFC of adult (P56–70) *Arid1b*[+/−] mice, as shown by resting membrane potential (RMP), action potential (AP) threshold, current-firing curve, and input resistance (IR). (n = 14, 5 [WT-RMP], 15, 5 [HT-RMP], 13, 5 [WT-AP threshold], 15, 5 [HT-AP threshold], 14, 5 [WT-current-firing], 15, 5 [HT-current-firing], 14, 5 [WT-IR], 15, 5 [HT-IR], two-way ANOVA [Firing rate, IR], two-tailed Student's t-test [AP threshold, RMP]). Graphical data are presented as means ± SEM (*p < 0.05, **p < 0.01, ***p < 0.001, ns, not significant). Source data are provided as a Source Data file.

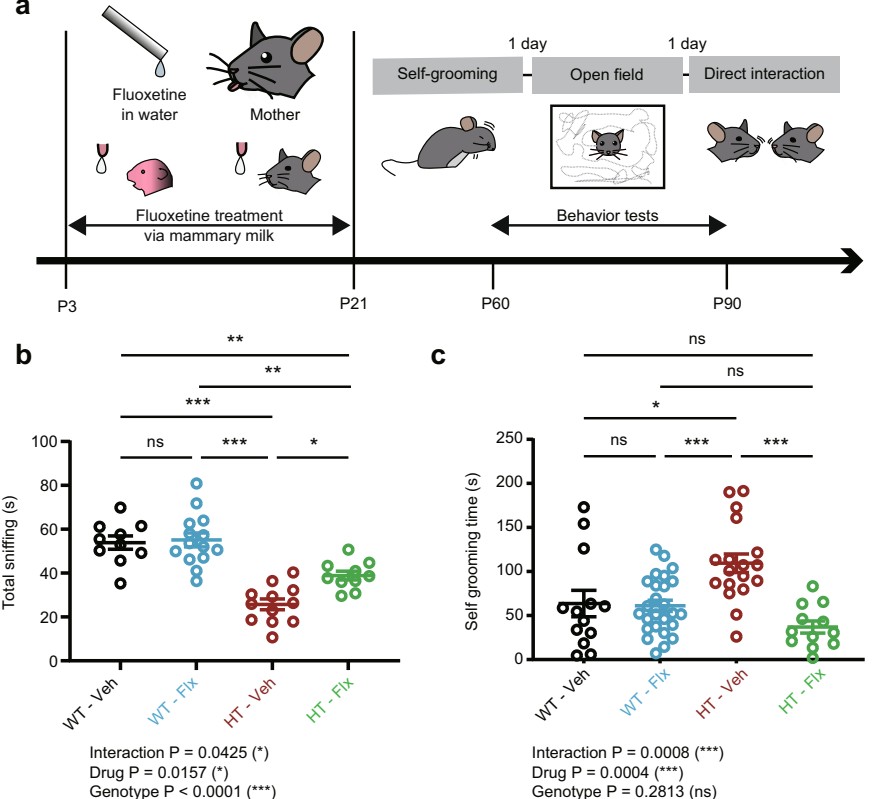

**Fig. 4 | Early, chronic fluoxetine treatment prevents social and repetitive behavioral deficits in adult *Arid1b*+/− mice. a** Schematic diagram showing early, chronic fluoxetine treatment (P3–21) through mother's milk and tests of its long-lasting effects on self-grooming, open-field activity, and direct social interaction in adulthood (P60–90). **b** Early, chronic fluoxetine prevents social defects in adult *Arid1b*+/− mice, measured using the direct/dyadic social-interaction test (as shown by time spent sniffing the social target) without affecting social interactions in WT mice. (*n* = 10 mouse pairs [WT-Veh], 14 [WT-Flx], 12 [HT-Veh], 10 [HT-Flx], two-way ANOVA with Tukey's test, [WT-Veh vs HT-Veh: *p* < 0.0001, WT-Veh vs WT-Flx:

*p* = 0.9899, WT-Veh vs HT-Flx: *p* = 0.0058, HT-Veh vs WT-Flx: *p* < 0.0001, HT-Veh vs HT-Flx: *p* = 0.0132, WT-Flx vs HT-Flx: *p* = 0.0010]). **c** Early, chronic fluoxetine prevents excessive self-grooming in adult *Arid1b*+/− mice (as shown by time spent self-grooming) without affecting grooming in WT mice. (*n* = 13 mice [WT-Veh], 26 [WT-Flx], 18 [HT-Veh], 12 [HT-Flx], two-way ANOVA with Tukey's test, [WT-Veh vs HT-Veh: *p* = 0.0119, WT-Veh vs WT-Flx: *p* = 0.9974, WT-Veh vs HT-Flx: *p* = 0.3400, HT-Veh vs WT-Flx: *p* = 0.0010, HT-Veh vs HT-Flx: *p* < 0.0001, WT-Flx vs HT-Flx: *p* = 0.3099]). Graphical data are presented as means ± SEM (**p* < 0.05, ***p* < 0.01, ****p* < 0.001, ns, not significant). Source data are provided as a Source Data file.

visualization (Fig. 6a; Supplementary Dataset 4). This pattern contrasts with—in fact, is opposite to—the downregulation of ribosome/mitochondria-related genes observed in naïve *Arid1b*+/− mice at P10 (Fig. 2a), suggestive of age-dependent, gradual—and opposite—changes in ribosome/mitochondria-related functions.

In addition, Veh-Arid/WT whole-brain transcripts at P120 displayed both ASD-like and reverse-ASD enrichment patterns. ASD-like changes were exemplified by negative enrichment for SFARI (all and high-confidence), DeNovoMiss, and DeNovoVariant (not FMRP target) gene sets, and also negative enrichment for oligodendrocytes (Fig. 6c, d). Reverse-ASD patterns were characterized by negative enrichment for the DEG_Up (but not Co-Exp_Up) gene set and positive enrichment for the Co-Exp_Down (but not DEG_Down) gene set. These mixed patterns at P120 differ from the strong reverse-ASD pattern observed at P10 (Fig. 2c, d), suggestive of an age-dependent progressive change from a strong reverse-ASD to a mixed ASD-like and reverse-ASD pattern.

In the mPFC, strong differences between P10 and P120 transcriptomes were also observed (Supplementary Fig. 12; Supplementary Dataset 4). For instance, Veh-Arid/WT mPFC transcripts displayed positive enrichment for synapse-related gene sets and negative enrichment for mitochondria/ribosome-related gene sets as well as reverse-ASD patterns such as positive enrichments for ASD-risk and excitatory neuron-related gene sets (Supplementary Fig. 12a, c). These patterns are largely opposite those observed in the P10 Arid/WT mPFC transcripts, further suggesting age-dependent progressive changes.

A comparison of whole-brain transcripts from fluoxetine- or vehicle-treated *Arid1b*+/− mice (Arid1b-Flx/Veh transcripts) indicated positive enrichment for synapse- and ribosome/mitochondria-related gene sets, as further supported by EnrichmentMap (Fig. 6b). In addition, Arid1b-Flx/Veh transcripts were negatively enriched for chromatin-related gene sets (Fig. 6b).

For ASD-related and cell-type-specific gene sets, Arid1b-Flx/Veh whole-brain transcripts displayed two features that markedly differed from the ASD-related pattern observed in Veh-Arid1b/WT transcripts and which favor reverse-ASD directions, namely positive enrichment for FMRP targets and oligodendrocytes (Fig. 6c, d).

In the mPFC, Arid1b-Flx/Veh transcripts displayed negative enrichment for ribosome/mitochondria-related gene sets (Supplementary Fig. 12b), which differs from the positive enrichments for these gene sets in whole-brain Arid1b-Flx/Veh transcripts (Fig. 6b). However, Arid1b-Flx/Veh mPFC transcripts displayed features of reverse-ASD patterns such as positive enrichment for FMRP target gene sets (Supplementary Fig. 12c), similar to that observed in whole-brain Arid1b-Flx/Veh whole-brain transcripts (Fig. 6c). In addition, mPFC Arid1b-Flx/Veh transcripts were negatively enriched for astrocyte- and microglia-related gene sets (Supplementary Fig. 12d), dissimilar to the positive enrichments for oligodendrocyte-related gene sets in whole-brain Arid1b-Flx/Veh transcripts (Fig. 6d).

These results collectively suggest that early, chronic fluoxetine treatment induces, in the whole-brain, upregulation of synapse/

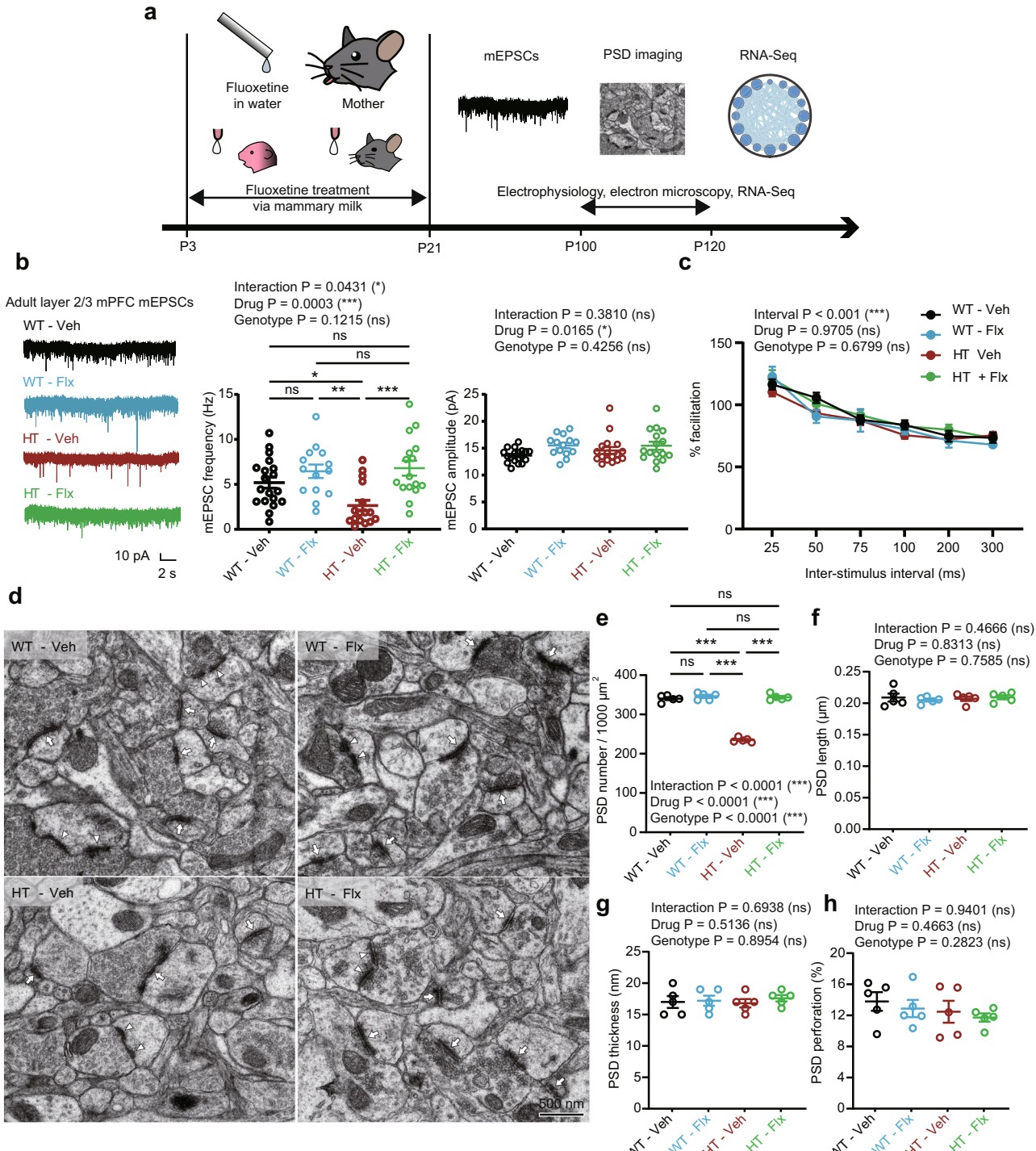

**Fig. 5 | Early, chronic fluoxetine treatment prevents excitatory synaptic deficits appearing in adult *Arid1b*+/− mice. a** Schematic diagram showing early, chronic fluoxetine treatment (P3–21) through mother's milk and prevention of excitatory synaptic deficits at P100, as determined by measuring mEPSCs. **b** Early, chronic fluoxetine prevents the decrease in mEPSC frequency in layer 2/3 pyramidal neurons in the prelimbic region of the mPFC of adult *Arid1b*+/− mice without affecting mEPSC frequency in WT mice. (*n* = 19 neurons from 4 mice [WT-Veh], 14, 3 [WT-Flx], 16, 3 [HT-Veh], and 16, 3 [HT-Flx], two-way ANOVA with Tukey's test, [frequency, WT-Veh vs HT-Veh: *p* = 0.0462, WT-Veh vs WT-Flx: *p* = 0.5802, WT-Veh vs HT-Flx: *p* = 0.3373, HT-Veh vs WT-Flx: *p* = 0.0025, HT-Veh vs HT-Flx: *p* = 0.0005, WT-Flx vs HT-Flx: *p* = 0.9866]). **c** Early, chronic fluoxetine does not affect paired-pulse facilitation in layer 2/3 pyramidal neurons in the prelimbic region of the mPFC of adult

*Arid1b*+/− mice. (*n* = 17 neurons from 3 mice [WT-Veh], 11, 3 [WT-Flx], 16, 3 [HT-Veh], and 12, 3 [HT-Flx], three-way ANOVA, [Interaction: *p* = 0.5047]). **d–h** Early, chronic fluoxetine prevents the decrease in excitatory synapse density in layer 2/3 pyramidal neurons in the prelimbic region of the mPFC of adult *Arid1b*+/− mice without affecting excitatory synapse density in WT mice, as shown by electron microscopic analysis of PSD density and morphology. (*n* = 5 mice [WT-Veh], 5 [WT-Flx], 5 [HT-Veh], and 5 [HT-Flx], two-way ANOVA with Tukey's test, [density, WT-Veh vs HT-Veh: *p* < 0.0001, WT-Veh vs WT-Flx: *p* = 0.6609, WT-Veh vs HT-Flx: *p* = 0.8977, HT-Veh vs WT-Flx: *p* < 0.0001, HT-Veh vs HT-Flx: *p* < 0.0001, WT-Flx vs HT-Flx: *p* = 0.9661]). Graphical data are presented as means ± SEM (**p* < 0.05, ***p* < 0.01, ****p* < 0.001, ns, not significant). Source data are provided as a Source Data file.

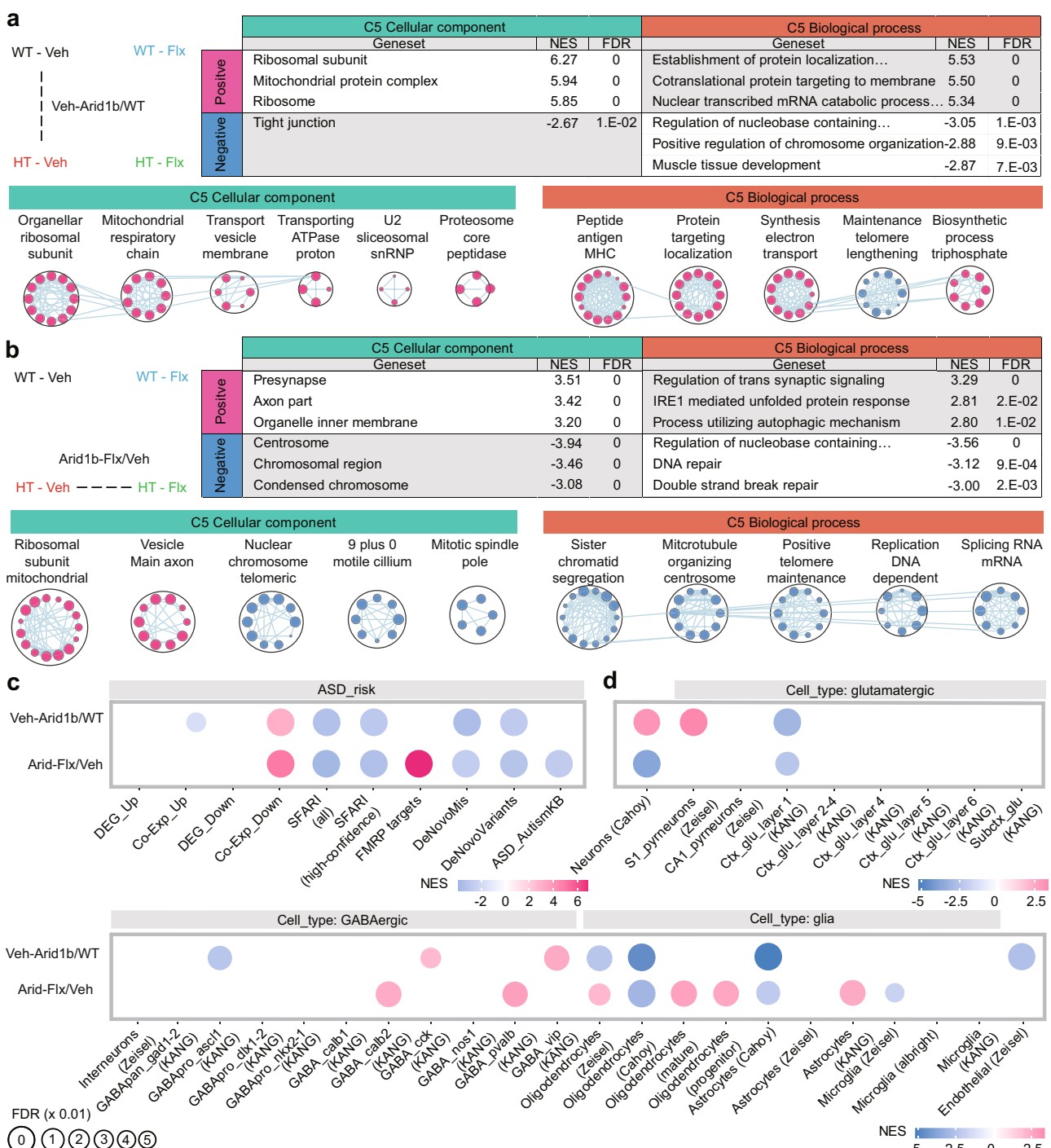

**Fig. 6 | Early, chronic fluoxetine treatment prevents aberrant transcriptomic changes in the whole brain of adult *Arid1b*+/− mice. a** Baseline differences in whole-brain Arid1b/WT transcripts associated with biological functions in early vehicle-treated (P3–21) mice, measured in whole-brain lysates at P120 (Veh-HT/WT transcripts), as shown by the top-three enriched gene-set lists from GSEA using C5-CC and C5-BP gene sets and EnrichmentMap integration and visualization of enriched gene sets. NES normalized enrichment score; FDR false discovery rate. (*n* = 5 mice [WT-Veh], 5 [HT-Veh], 5 [WT-Mem], 5 [HT-Mem]). **b** Transcriptomic changes associated with biological functions induced by early, chronic fluoxetine treatment (P3–21) in the whole brain of adult *Arid1b*+/− mice, measured at P120 (Arid1b-Flx/Veh transcripts), as shown by the top-three enriched gene-set lists and EnrichmentMap integration/visualization. (*n* = 5 mice [WT-Veh], 5 [HT-Veh], 5 [WT-Mem], 5 [HT-Mem]). **c, d** ASD-related and cell-type–specific whole-brain transcriptomic changes in early vehicle-treated *Arid1b*+/− and WT mice (Veh-Arid1b/WT transcripts) and early fluoxetine/vehicle-treated *Arid1b*+/− mice (Arid1b-Flx/Veh transcripts), as shown by enrichment patterns for ASD-related/risk gene sets and cell-type-specific gene sets. (*n* = 5 mice [WT-Veh], 5 [HT-Veh], 5 [WT-Mem], 5 [HT-Mem]).

ribosome/mitochondria-related genes and downregulation of chromatin-related genes, and promotes a reverse-ASD pattern through transcriptomic upregulation of FMRP target- and oligodendrocyte-related genes. In addition, early chronic fluoxetine treatment induces, in the mPFC, downregulation of ribosome/mitochondria-related genes and promotes a reverse-ASD pattern through upregulation of FMRP target genes and downregulation of astrocyte/microglia-related genes.

**Early, chronic fluoxetine treatment normalizes HDAC4- and MEF2A-dependent transcription in *Arid1b*$^{+/-}$ mice**

To better understand how early fluoxetine treatment restores excitatory synaptic density and transmission in adult *Arid1b*$^{+/-}$ mice, we sought to identify molecular and cellular mechanisms by performing unbiased RNA-Seq. One notable change shared in the whole-brain and mPFC transcriptome of adult *Arid1b*$^{+/-}$ mice treated early with fluoxetine was strong positive enrichment of the FMRP target gene set (Fig. 6c; Supplementary Fig. 12).

This enrichment was most strongly attributable to the *Hdac4* gene encoding histone deacetylate 4 protein (HDAC4) (Fig. 7a), which acts as a co-repressor of the ASD-related transcription factor MEF2A[46] which in turn upregulates the negative synaptic regulators SynGAP1 and Arc[47–49] to suppress excitatory synaptic and cognitive function[50–57]. This suggests the possibility that HDAC4 level/activity is decreased in *Arid1b*$^{+/-}$ mice, leading to insufficient repression of MEF2A and thus excessive SynGAP/Arc expression and synaptic suppression, and that early fluoxetine treatment of *Arid1b*$^{+/-}$ mice may reverse these changes (Fig. 7b).

In support of this hypothesis, HDAC4 levels were decreased in the whole brain of *Arid1b*$^{+/-}$ mice relative to WT mice, and were rescued by early fluoxetine treatment, which had no effect in WT mice (Fig. 7c). Similar, albeit less striking, results were obtained in nucleus-enriched fractions. The comparable changes in total and P1 levels (Fig. 7c) suggest that decreased amounts of HDAC4 protein suppress HDAC4 nuclear translocation.

In addition, although total MEF2A levels in *Arid1b*$^{+/-}$ mice were unchanged, there was a moderate decrease in the levels of S408-phosphorylated MEF2A, representing nuclear-translocated protein, that was rescued by fluoxetine treatment (Fig. 7d). WT mice showed no such decrease or fluoxetine-dependent rescue. Moreover, levels of the MEF2A downstream targets, SynGAP1 and Arc, were concomitantly increased in the *Arid1b*$^{+/-}$ mice and were rescued by early fluoxetine treatment (Fig. 7e).

In the mutant mPFC, decreased HDAC4 levels and its fluoxetine-dependent rescue were observed (Supplementary Fig. 13a), similar to the results from the whole brain (Fig. 7c). In addition, increased SynGAP1 levels and its fluoxetine-dependent rescue were observed (Supplementary Fig. 13b), similar to the results from the whole brain (Fig. 7e). No detectable differences were observed for Arc levels (Supplementary Fig. 13c), suggestive of brain region-specific changes. Total and phosphorylation levels of MEF2A could not be detected in the mPFC for small amounts of samples relative to those from the whole brain. Transcript levels for HDAC4, MEF2A, SynGAP1, and Arc in the whole brain and mPFC at ~P120 did not show significant changes in *Arid1b*$^{+/-}$ mice, although the directions of the changes tended to be similar to those observed in the proteins, i.e., HDAC4 (Supplementary Fig. 13d).

In contrast to these alterations in HDAC4, MEF2A, SynGAP1, and Arc proteins, changes in phosphorylation of β-catenin at S552 or S675, which are known to promote β-catenin nuclear translocation and subsequent transcriptional changes[58,59]—and were previously reported to occur in another *Arid1b*-mutant mouse line[24]—were not observed in our naïve or fluoxetine-treated *Arid1b*$^{+/-}$ mice (Supplementary Fig. 14).

These results collectively suggest that *Arid1b* haploinsufficiency leads to decreased total and nuclear levels of HDAC4, decreased activity of the HDAC4-MEF2A transcription repressor complex, and increased expression of SynGAP1/Arc, and further that early postnatal fluoxetine treatment reverses these changes.

**Brain 5-HT levels and serotonergic neuronal *Arid1b* expression minimally affect *Arid1b*$^{+/-}$ phenotypes**

Because fluoxetine inhibits serotonin (5-HT) reuptake, the long-lasting effects of early fluoxetine treatment in *Arid1b*$^{+/-}$ mice may involve rescue of decreased brain levels of serotonin in these mice.

However, whole-brain 5-HT levels were not decreased in the brains of *Arid1b*$^{+/-}$ mice and were not altered by early fluoxetine treatment (P3–10) in mutant or WT mice (Fig. 8a). In addition, forced-swim tests revealed no detectable depression-like behavior—a serotonin-related attribute—in naïve *Arid1b*$^{+/-}$ mice (Fig. 8b).

Because it is also possible that ARID1B expression in 5-HT–expressing neurons is important, we restricted *Arid1b* deletion to serotonergic neurons in the raphe nucleus, where *Arid1b* is expressed (Fig. 8c), by crossing an *Arid1b*$^{+/flox}$ line, which we additionally generated in the present study, with a Sert-Cre line (*Sert-Cre; Arid1b*$^{+/flox}$) (Fig. 8d, e). However, direct social interaction and self-grooming were unaltered in *Sert-Cre;Arid1b*$^{+/flox}$ mice compared with control mice (*Arid1b*$^{+/flox}$ mice without Cre recombination) (Fig. 8f, g).

These results collectively suggest that neither decreased brain 5-HT levels nor *Arid1b* deletion in serotonergic neurons contributes to the phenotypic deficits in *Arid1b*$^{+/-}$ mice.

## Discussion

In this study, we demonstrated that early, chronic fluoxetine treatment of *Arid1b*$^{+/-}$ mice prevents transcriptomic, synaptic, and behavioral deficits in adults, suggesting that early postnatal treatment is critical for long-lasting prevention of ARID1B-related phenotypes. In addition, we identified excitatory synaptic suppression as a pathophysiology associated with *Arid1b* deficiency in mice and suggested early postnatal, chronic serotonin receptor modulation as a strategy for correcting *Arid1b*-related phenotypes through synapse-related transcriptional reprogramming.

Increasing evidence underscores the importance of identifying early-stage mechanistic deviations in neurodevelopmental disorders, including ASD, and taking advantage of the flexible and modifiable nature of the nervous system to correct them during these critical early postnatal periods so as to attain long-lasting effects[9,10,60]. However, despite increasing support for this concept, efforts to identify key early mechanisms operating at molecular and synaptic levels and correct them for long-lasting effects have thus far fallen short. The present results identify decreased excitatory synaptic density and transmission that persists from juvenile to adult stages as a key ARID1B-associated pathophysiology. In addition, early, chronic fluoxetine treatment (P3–21) has long-lasting effects, preventing adult-stage synaptic and behavioral deficits through mechanisms including HDAC4/MEF2A-related transcriptional reprogramming.

The decreased excitatory synaptic density and transmission observed in our *Arid1b*$^{+/-}$ mice contrasts with the decrease in inhibitory synaptic transmission previously reported in an *Arid1b*-deficient mouse line[24]. This discrepancy could be attributable to differences in the strategy for deleting the same exon 5 in the two different mouse lines (TT insertion to exon 5 in our study vs. exon 5 deletion in the previous study) or the distinct genetic backgrounds of mice (C57BL6/N vs. C57BL6/J), which are known to substantially alter synaptic and behavioral phenotypes through mechanisms involving the Cyfip2 protein[61]. Other possible explanations for the discrepancy include different conditions for electrophysiological measurements (e.g., holding potentials and/or internal/external solutions) or animal breeding/handling conditions. Although further details remain to be clarified, our results and previous reports suggest that, at minimum, an *Arid1b* haploinsufficiency leads to synaptic dysfunctions, despite their evident occurrence at distinct (excitatory and inhibitory) synapses. It should be noted, however, that the changes reported in both studies would similarly disrupt the balance between synaptic excitation and inhibition, a key mechanism underlying ASD[62–64].

Early postnatal fluoxetine treatment prevents adult-stage synaptic deficits through mechanisms that include upregulation of FMRP targets and HDAC4/MEF2A-dependent transcriptional reprogramming. Specifically, early fluoxetine treatment increases the expression of FRMP target genes in both the whole brain and mPFC of *Arid1b*$^{+/-}$ mice

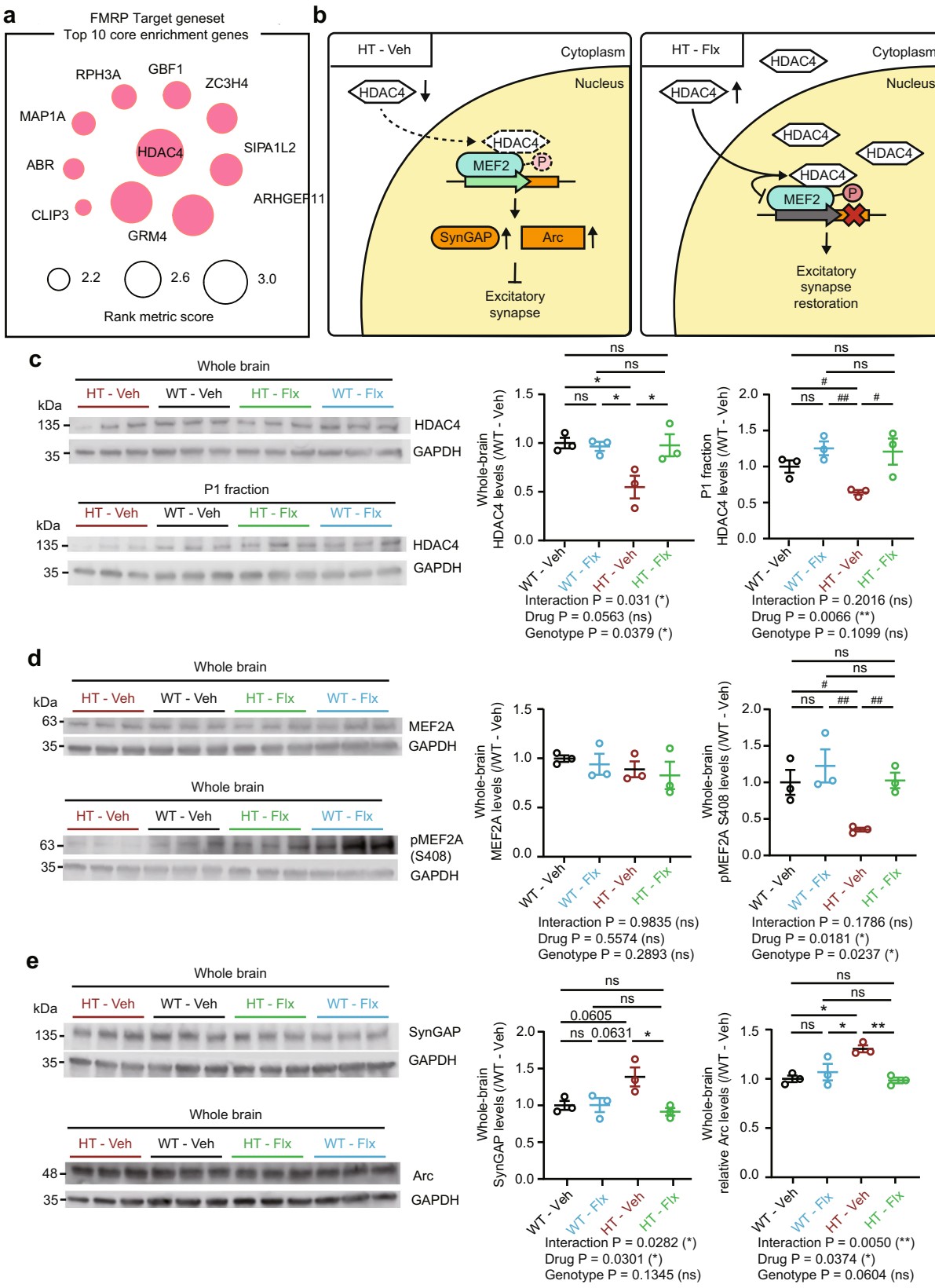

(Fig. 6c and Supplementary Fig. 12c). FMRP targets, together with risk genes for ASD and intellectual disability, are known to regulate synaptic functions[65–67]. In addition, early fluoxetine treatment increases the levels of the HDAC4 protein, a key FMRP target, in the whole brain and mPFC of *Arid1b*[+/−] mice (Fig. 7c and Supplementary Fig. 13a). This promotes the nuclear localization of HDAC4 (Fig. 7c) and HDAC4-

dependent inhibition of the MEF2A transcription factor (Fig. 7d), known to inhibit excitatory synapses[50–57,68] by promoting the expression of SynGAP1 (inhibitor of Ras small GTPases) and Arc (promoter of metabotropic glutamate receptor/mGluR-mediated long-term depression)[47–49]. Indeed, SynGAP1 levels are decreased in early fluoxetine-treated whole brain and mPFC of *Arid1b*[+/−] mice (Fig. 7e and

**Fig. 7 | Early, chronic fluoxetine treatment normalizes HDAC4- and MEF2A-dependent transcription in adult *Arid1b*⁺/⁻ mice. a** Genes that strongly contribute to positive enrichment of Arid1b-Flx/Veh transcripts in the FMRP target gene set (Fig. 6c). **b** The working hypothesis in the left panel suggests that a decrease in the total levels of HDAC4 in *Arid1b*⁺/⁻ mice may lead to decreases in the nuclear levels of HDAC4, insufficient repression of the transcription factor MEF2A (as shown by insufficient phosphorylation), and resulting excessive expression of SynGAP1/Arc (negative synaptic regulators) and suppression of excitatory synapses. The right panel suggests that these changes in *Arid1b*⁺/⁻ mice are reversed by early postnatal chronic fluoxetine treatment. **c** Total and nuclear (P1 fraction) levels of HDAC4 protein in adult (P120) WT and *Arid1b*⁺/⁻ mice treated early (P3–21) with vehicle or fluoxetine. ($n = 3$ [WT-Veh], 3 [WT-Flx], 3 [HT-Veh], and 3 [HT-Flx], two-way ANOVA with Tukey's test and two-tailed Student's t-test; the positive drug effect in the ANOVA led us to attempt two-tailed Student's t-tests for multiple comparisons (indicated by # signs), [Whole-brain, WT-Veh vs HT-Veh: $p = 0.0285$, WT-Veh vs WT-Flx: $p = 0.9925$, WT-Veh vs HT-Flx: $p = 0.9978$, HT-Veh vs WT-Flx: $p = 0.0416$, HT-Veh vs HT-Flx: $p = 0.0366$, WT-Flx vs HT-Flx: $p = 0.9997$], [P1 fraction, WT-Veh vs HT-

Veh: $p = 0.0173$, WT-Veh vs WT-Flx: $p = 0.1209$, WT-Veh vs HT-Flx: $p = 0.3610$, HT-Veh vs WT-Flx: $p = 0.0038$, HT-Veh vs HT-Flx: $p = 0.0375$, WT-Flx vs HT-Flx: $p = 0.8347$]). **d** Total and S408-phosphorylated levels of MEF2A in adult (P120) WT and *Arid1b*⁺/⁻ mice treated early (P3–21) with vehicle or fluoxetine. ($n = 3$ [WT-Veh], 3 [WT-Flx], 3 [HT-Veh], and 3 [HT-Flx], two-way ANOVA and two-tailed Student's t-test, [pMEF2A, WT-Veh vs HT-Veh: $p = 0.0199$, WT-Veh vs WT-Flx: $p = 0.8571$, WT-Veh vs HT-Flx: $p = 0.9018$, HT-Veh vs WT-Flx: $p = 0.0093$, HT-Veh vs HT-Flx: $p = 0.0035$, WT-Flx vs HT-Flx: $p = 0.7051$]). **e** Total levels of SynGAP1 and Arc proteins in adult (P120) WT and *Arid1b*⁺/⁻ mice treated early with (P3–21) vehicle or fluoxetine. ($n = 3$ [WT-Veh], 3 [WT-Flx], 3 [HT-Veh], and 3 [HT-Flx], two-way ANOVA with Tukey's test, [SynGAP, WT-Veh vs HT-Veh: $p = 0.0605$, WT-Veh vs WT-Flx: $p > 0.9999$, WT-Veh vs HT-Flx: $p = 0.9007$, HT-Veh vs WT-Flx: $p = 0.0631$, HT-Veh vs HT-Flx: $p = 0.0233$, WT-Flx vs HT-Flx: $p = 0.8893$], [Arc, WT-Veh vs HT-Veh: $p = 0.0118$, WT-Veh vs WT-Flx: $p = 0.7801$, WT-Veh vs HT-Flx: $p = 0.9961$, HT-Veh vs WT-Flx: $p = 0.0089$, WT-Flx vs HT-Flx: $p = 0.6623$]). Graphical data are presented as means ± SEM (*$p < 0.05$, **$p < 0.01$, ***$p < 0.001$, ns, not significant). Source data are provided as a Source Data file.

Supplementary Fig. 13b). In line with this hypothesis, HDAC4/SynGAP1 and MEF2A are associated with ASD[32] and schizophrenia[69], respectively, and HDAC4 modulation shows promise in treating neurodegenerative disorders[70]. However, it should be noted that ARID1B-related pathologies may be associated with larger transcriptional changes surrounding HDAC4/MEF2A because HDAC4 acts on many other nuclear/cytoplasmic proteins (in addition to MEF2A)[70,71], and MEF2A-dependent suppression of excitatory synapses involves multiple upstream regulators (in addition to NMDA receptors) and downstream targets (in addition to SynGAP1 and Arc)[52,72].

Our results strongly suggest that early postnatal serotonin receptor modulation has beneficial effects in *Arid1b*⁺/⁻ mice. Serotonin regulates brain development and synaptic plasticity[73], and its dysfunction has been implicated in various neurodevelopmental disorders, including ASD[73], and identified in mouse models of ASD[45,74,75]. In addition, serotonin receptor modulation by fluoxetine treatment or using genetic approaches mitigates ASD-related mouse phenotypes[74,75]. However, with the exception of a single recent report[45], to the best of our knowledge, these studies have mainly targeted adult mice rather than newborn/juvenile mice at critical developmental periods. This one exception, however, further differs from our study in that the major deviating mechanism under study was a decrease in brain serotonin levels and the focus was on correction of serotonin levels and phenotypic rescue. Our results from both *Arid1b*⁺/⁻ mice (global KO) and *Arid1b*-conditional-KO mice (serotonergic neurons) suggest that altered brain serotonin levels are not likely involved in rescuing the mouse phenotypes. Instead, early, chronic fluoxetine treatment of *Arid1b*-deficient mice may activate 5-HT receptors and downstream signaling pathways to modulate transcriptional reprograming involving HDAC4/MEF2A. In line with this, 5-HT receptor activation has been shown to regulate transcriptional programs through epigenetic mechanisms, including DNA methylation, histone modification, and microRNAs[73]. However, how early fluoxetine treatment elevates HDAC4 levels remains unclear, although chromatin-related gene expressions are strongly decreased in fluoxetine-treated *Arid1b*⁺/⁻ mice (Fig. 6b), suggesting that chromatin remodeling-related transcriptional changes may be involved.

Lastly, and more broadly, many previous studies on mouse models of ASD have reported excitatory synaptic dysfunctions, including decreased NMDA-, AMPA-, and mGluR-related functions, which frequently are rescued by acute pharmacological treatment at adult stages[2,27,67,76,77]. However, metabolic degradation of these drugs and the largely inflexible nature of the nervous system at adult stages would inevitably end in a waning effect of such treatments. Inducing synaptic transcriptional reprogramming through early, chronic postnatal fluoxetine treatment during critical periods may be a way to address these limitations and achieve long-lasting effects.

In summary, our results suggest that *Arid1b* haploinsufficiency leads to excitatory, but not inhibitory, synaptic deficits and ASD-like behavioral deficits that are responsive to early, chronic fluoxetine treatment, which has long-lasting preventive effects on ASD-related phenotypes in adults. Our results suggest early serotonin receptor modulation-dependent prevention of adult-stage ARID1B-related phenotypes through HDAC4/MEF2A-dependent transcriptional reprograming.

## Methods
### Animals
Experimental procedures using mice (Mus musculus) were approved by the Committee on Animal Research at KAIST (KA2016-33 and KA2020-51) and performed in compliance with all relevant ethical regulations. Mice were maintained according to the requirements of Animal Research at KAIST and fed ad libitum and housed under a 13:00-01:00 dark/light cycle.

*Arid1b*⁺/⁻ mice in which exon 5 of the *Arid1b* gene was targeted by CRISPR/Cas9 (C57BL/6N-Arid1b^emlTcp) were generated as a part of the NorCOMM2 project (TCPC317) funded by Genome Canada and the Ontario Genomics Institute (OGI-051) at the Toronto Center for Phenogenomics, and were obtained from the Canadian Mouse Mutant Repository. Targeting of *Arid1b* exon 5 employed Cas9 nickase (D10A) and single-guide RNAs with spacer sequences CTGCTTAGCAAGT-TACCACT and GCCTGATACAGCACTTACAT, for targeting the 5′ side of exon OTTMUSE00000314956 (exon 5), and ACACTAAAGGGG TTGCTTTC and CTTGTAATCCCCCTGTAGTA, for targeting the 3′ side, resulting in deletion of Chr17 from 5242523 to 5243410 with insertion of "TT". All mice used in the present study were generated by crossing male *Arid1b*⁺/⁻ mice with female WT C57BL/6N mice. Pups were weaned at P21, and 3–6 mice were grouped in each cage. Mice in the same genetic background were co-housed after weaning.

For conditional *Arid1b* knockout in serotonergic neurons, transgenic mice harboring a cassette containing the *Ardi1b* gene flanked by loxP and lacZ sites flanked by FRT sites were obtained from EUCOMM. This Arid1b^tmIa(EUCOMM)Hmgu mouse line was crossed with *Protamine-Flp* mice to remove the cassette and produce *Arid1b*^fl/+ mice, which were then backcrossed with C57BL/6J mice to change the genetic background. *Sert-Cre; Arid1b*^fl/+ mice were generated by crossing Sert-Cre female mice from Jackson Laboratory (B6.129(Cg)-*Slc6a4*^tml(cre)Xz/J) with *Arid1b*^fl/fl male mice. Thy1-EGFP mice were from Jackson laboratory (Tg(Thy1-EGFP)MJrs/J).

For PCR genotyping, the following primers were used. Arid1b HT, primer F1: CATTACAGTGTCCTCTCCCATCTTG, R1: GAAAGAGAAAG CGGGTGTTCATAC, R2: CGGTGTGTGACTGTGATCATAGATG; Arid1b^fl/+, F: GTGTCCTCTCCCATCTTGCCTGCCTTCTCT, R: TCACAAATGGCTT GACGGTTCCCCTT; B6.129(Cg)-*Slc6a4*^tml(cre)Xz/J, F1: GAGCTCTCAGT

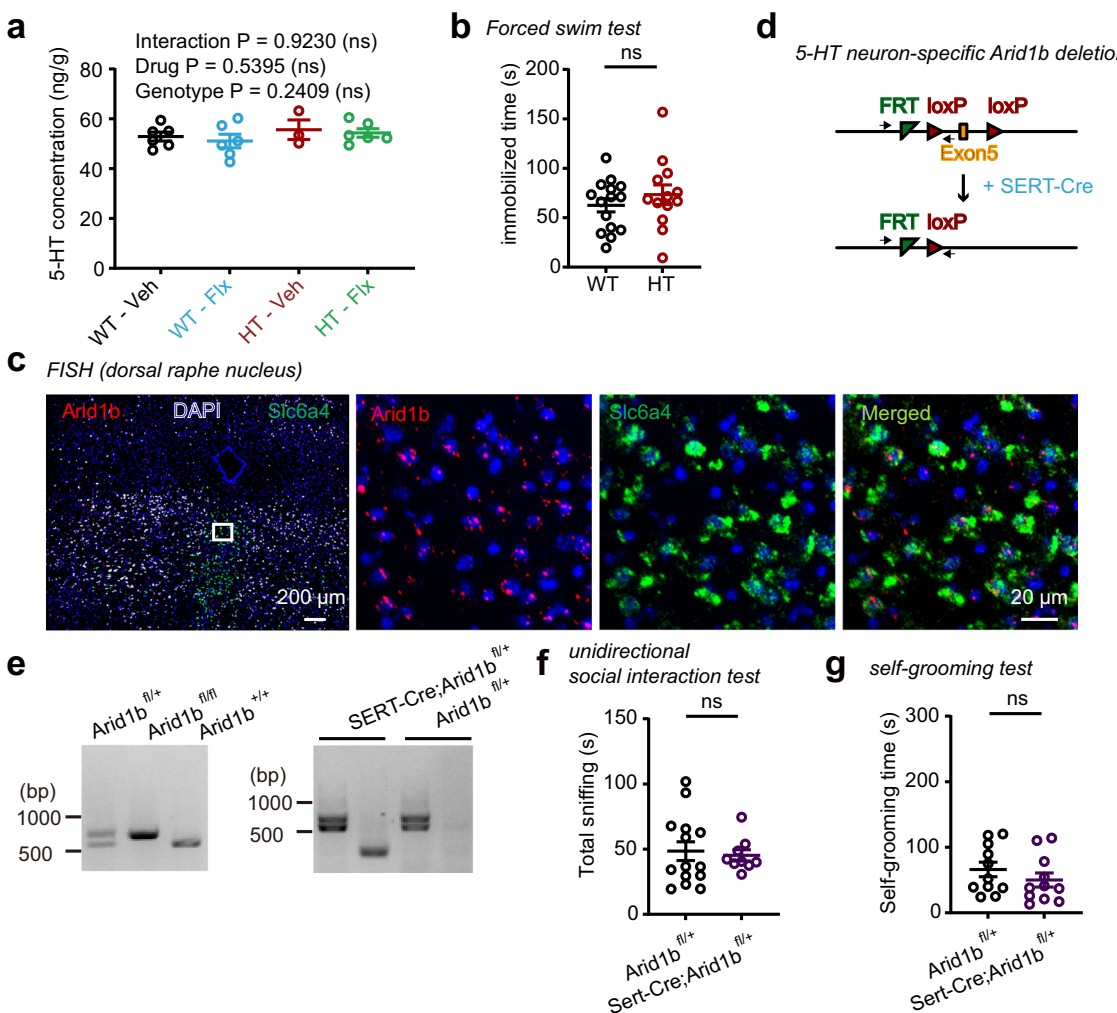

**Fig. 8 | Brain 5-HT levels and serotonergic neuronal *Arid1b* expression minimally affect *Arid1b*[+/–] phenotypes. a** Comparable 5-HT concentrations in whole brains of vehicle- or fluoxetine-treated (P3–10) WT and *Arid1b*[+/–] mice, determined at P10 by LC-MS/MS. (n = 6 mice [WT-Veh], 3 [HT-Veh], 6 [WT-Flx], 6 [HT-Flx], two-way ANOVA). **b** Comparable levels of depression-like behavior in adult (P80) WT and *Arid1b*[+/–] mice, as indicated by immobile time in the forced-swim test. (n = 15 mice [WT], 13 [HT], two-tailed Student's t-test, [p = 0.3632]). **c** Fluorescence in situ hybridization (FISH) showing expression of *Arid1b* in serotonergic neurons positive for *Slc6a4*, encoding the serotonin transporter SERT. Similar results were obtained from two independent experiments. **d** Schema for the generation of mice with conditional deletion of exon 5 of the *Arid1b* gene restricted to serotonergic neurons

(*Sert-Cre; Arid1b*[+/fl]) by crossing *Arid1b*[+/fl] and *Sert-Cre* mouse lines. **e** PCR genotyping of *Arid1b*[+/fl], *Arid1b*[fl/fl], and *Sert-Cre;Arid1b*[+/fl] mice. Similar results were obtained from three independent experiments. **f** Normal social interaction in adult (P80) male *Sert-Cre; Arid1b*[+/fl] mice in the direct social-interaction test using a stranger mouse with a C3H genetic background, compared with control *Arid1b*[+/fl] mice, as shown by time spent sniffing the stranger. (n = 14 mice [Arid1b[+/fl]], 9 [Sert-Cre; Arid1b[+/fl]], two-tailed Mann–Whitney test). **g** Normal self-grooming in adult (P80) male *Sert-Cre;Arid1b*[+/fl] mice compared with *Arid1b*[+/fl] control mice, as shown by total time spent self-grooming. (n = 11 [Arid1b[+/fl]], 11 [Sert-Cre;Arid1b[+/fl]], two-tailed Student's t-test). Graphical data are presented as means ± SEM (*p < 0.05, **p < 0.01, ***p < 0.001, ns, not significant). Source data are provided as a Source Data file.

---

CTTGTCTCCA, R1: GAGTGTGGCGCTTCATCC, R2: AGGCAAATTTTGG TGTACGG, F2: GTGTTGCCGCGCCATCTG, R2: CACCATTGCCCCTG TTTCACTATC; Tg(Thy1-EGFP)MJrs/J, F1: ACAGACACACACCCAGGA CA, R1: CGGTGGTGCAGATGAACTT

## Drug administration

Fluoxetine hydrochloride was administrated as previously described[45]. Briefly, fluoxetine was indirectly delivered to pups through mammary milk by providing dams with fluoxetine hydrochloride (LKT Laboratories) at a concentration of 0.1 mg/ml in 0.2% saccharin-based drinking water. Dams in the vehicle group were fed 0.2% saccharin water without fluoxetine. Dams and pups were separated from sires in mating cages 3 days after birth for initiation of drug treatment. After 18 days (P21), mice were weaned and co-housed in cages containing 3–6 mice with the same genetic background. After weaning, mice were fed normal drinking water. Dams administered drugs were not returned to mating cages. We did not

directly measure the levels of fluoxetine and its metabolites in the sera of newborn mice, but a previous study employing the same drug treatment strategy has demonstrated that fluoxetine can reach the blood of newborn mice via mother's milk and reported detailed concentrations of serum fluoxetine and its metabolite during and after the treatment[45]. For adult-stage chronic treatment of fluoxetine, mice were treated with fluoxetine hydrochloride (LKT Laboratories) at a concentration of 0.1 mg/ml in 0.2% saccharin-containing drinking water for 18 days.

## Electrophysiology

Mice for electrophysiology were anesthetized using isoflurane (Terrell), and brain sections (300 μm, sagittal hippocampal slices and coronal mPFC slices) were prepared with using a vibratome (Leica VT1200) in ice-cold artificial sectional cerebrospinal fluid (sCSF) buffer containing (in mM) 212 Sucrose, 25 $NaHCO_3$, 10 D-glucose, 5 KCl, 2 Na-pyruvate, 1.25 L-ascorbic acid, 1.25 $NaH_2PO_4$, 3.5 $MgSO_4$, 0.5 $CaCl_2$

bubbled with 95% $O_2$ and 5% $CO_2$ gas. After sectioning, slices were recovered in chambers with artificial cerebrospinal fluid (ACSF) held at 32 °C containing (in mM) 125 NaCl, 25 $NaHCO_3$, 2.5 KCl, 1.25 $NaH_2PO_4$, 1.3 $MgCl_2$, and 2.5 $CaCl_2$ bubbled with 95% $O_2$ and 5% $CO_2$ gas for 30 min. Then, chamber was moved to room temperature and slices were recovered for 30 min while being bubbled with 95% $O_2$ and 5% $CO_2$ gas. Brain slices were moved to a recording chamber, perfused with circulating ACSF at 28 °C. For stimulations and recordings, borosilicate glass pipettes (Harvard Apparatus) were pulled using an electrode puller (Narishige).

For whole-cell recording, recording pipettes (2.5–3.5 MΩ) were filled with the folloiwng internal solutions; (1) EPSC experiments (in mM: 117 CsMeSO4, 10 TEA-Cl, 8 NaCl, 10 HEPES, 5 QX-314-Cl, 4 Mg-ATP, 0.3 Na-GTP, 10 EGTA with pH 7.3 and 285–300 mOsm), and (2) IPSC experiments (in mM: 115 $CsCl_2$, 10 TEA-Cl, 8 NaCl, 10 HEPES, 5 QX-314-Cl, 4 Mg-ATP, 0.3 Na-GTP, 10 EGTA with pH 7.3 and 285-300 mOsm). The response was filtered at 2 kHz and digitized at 10 kHz (Multiclamp 700B, Molecular Devices, and Digidata 1550, Molecular Devices). pClamp (10.1, Molecular Devices) was used to acquire data file. The acquired data were analyzed using Clampfit 10 (Molecular Devices). Series resistance was monitored by measuring the peak amplitude of the capacitance currents in response to hyperpolarizing step pulses (5 mV, 40 ms). Miniature currents were measured while holding the voltage at −70 mV. For miniature current recordings, tetrodotoxin (10 µM, Abcam) were added to ACSF to inhibit action potential firing. Additionally, for mEPSCs measurements, picrotoxin (100 µM, Abcam) was added to prevent inhibitory currents. For mIPSCs measurements, NBQX (100 µM, Tocris) and D-AP5 (100 µM, Tocris) were additionally added to ACSF to block AMPA and NMDA receptor-mediated currents, respectively.

To measure paired-pulse facilitation in mPFC layer 2/3 pyramidal neurons, picrotoxin (100 µM) was added to ACSF to prevent inhibitory currents. Paired pulses with 25, 50, 75, 100, 200, and 300 ms intervals were given. To measure the excitability of mPFC layer 2/3 pyramidal neurons, recording pipettes (2.5–3.5 MΩ) were filled with an internal solution containing (in mM: 137 K-gluconate, 5 KCl, 10 HEPES, 4 Mg-ATP, 0.5 Na-GTP, 10 Na-phosphocreatine, 0.2 EGTA with pH 7.2 and 280 mOsm). To inhibit postsynaptic response, picrotoxin (100 µM), NBQX (100 µM), D-AP5 (100 µM) were added to ACSF. After cell rupturing, the restring membrane potential was measured by current clamping and, after stabilization, was adjusted to −65 mV. The current input was increased from 0 to 200 pA, with 20 pA increments per sweep.

## Electron microscopy
Mice were deeply anesthetized with a mixture of ketamine (120 mg/kg) and xylazine (10 mg/kg) and were intracardially perfused with 10 ml of heparinized normal saline, followed by 50 ml of a freshly prepared fixative of 2.5% glutaraldehyde and 1% paraformaldehyde in 0.1 M phosphate buffer (PB; pH 7.4). Whole brains were removed and post fixed in the same fixative for 2 h and stored in PB (0.1 M, pH 7.4) overnight at 4 C. Brain sections (70 µm) were cut transversely on a vibratome. The sections were osmicated with 1% osmium tetroxide (in 0.1 MPB) for 1 h, dehydrated in graded alcohols, flat embedded in Durcupan ACM (Fluka), and cured for 48 h at 60 °C. Brain pieces containing the layer 2/3 region of the medial prefrontal cortex were cut out of the wafers and glued onto the plastic block by cyanoacrylate. Ultrathin sections were cut and mounted on Formvar-coated single slot grids. Sections were then stained with uranyl acetate and lead citrate, and examined with an electron microscope (Hitachi H-7500; Hitachi) at 80 kV accelerating voltage. Thirty-two micrographs representing 873.9 µm$^2$ neuropil regions in each mouse were photographed at a 30,000× and used for quantification. Quantification of the density, length, thickness, and perforation of the postsynaptic density were

performed by an experimenter blind to the genotype and drug treatment.

## Postembedding immunogold staining for GABA
Sections were immunostained for GABA by the postembedding immunogold method, as previously described[78] with some modifications. In brief, the grids were treated for 6 min in 1% periodic acid, to etch the resin, and for 8 min in 9% sodium periodate, to remove the osmium tetroxide, then washed in distilled water, transferred to Tris-buffered saline containing 0.1% Triton X-100 (TBST; pH 7.4) for 10 min, and incubated in 2% human serum albumin (HSA) in TBST for 10 min. The grids were then incubated with rabbit antiserum against GABA (1:150; AB5016; Chemicon) in TBST containing 2% HSA for 2 hrs at room temperature. To eliminate cross-reactivity, the diluted antiserum was preabsorbed overnight with glutaraldehyde (G)-conjugated glutamine (600 µM), as described previously[79]. After extensive rinsing in TBST, grids were incubated for 3 hrs in goat anti-rabbit IgG coupled to 15 nm gold particles (1:25 in TBST containing 0.05% polyethylene glycol; BioCell Co.). After a rinse in distilled water, the grids were counterstained with uranyl acetate and lead citrate, and examined with an electron microscope (Hitachi H-7500; Hitachi) at 80 kV accelerating voltage. To assess the immunoreactivity for GABA, gold particle density (number of gold particles per µm$^2$) of each GABA+ terminal was compared with gold particle density of terminals which contain round vesicles and make asymmetric synaptic contact with dendritic spines (background density). Terminals were considered GABA-immunopositive (+) if the gold particle density over the vesicle-containing areas was at least five times higher than background density.

## Quantitative analysis of inhibitory synapses
For quantification of inhibitory synapse, thirty-two electron micrographs representing 873.9 µm$^2$ neuropil regions in each mouse were taken at the magnification of 30,000. Number of GABA+ terminals from each three WT and $Arid1b^{+/-}$ mice were quantified by using ImageJ software. The measurements were all performed by an experimenter blind to the genotype.

## Behavioral assays
All behavioral tests were performed during light-off periods, and mice were habituated to the experimental rooms for more than 30 min. All subject mice were males except for pup USV tests, where a mix of males and females was used; female mice were not used in behavioral experiments because of the lack of male-female differences in clinical cases [17]. Unless specified otherwise, all data were analyzed using Ethovision XT 10.1 (Noldus) and were performed in a blinded manner. All behavioral tests, except for the dark condition in the open field test, were performed under a brightness of 100 lux.

## Repetitive self-grooming
For repetitive self-grooming test, subject mice were individually placed in new home cages without bedding. After 10 min habituation, time spent in self-grooming during 10 min was measured manually in a blind manner.

## Open-field test
Mice were individually placed in a custom-built box (33 × 33 × 22 cm). The subject mouse was allowed to freely explore the box for 1 h, and its activity was recorded. The brightness of the box was adjusted to 100 lux except for the 0 lux for the light-off open-field test. Distance traveled and time spent in the center region of the box were analyzed using Ethovision XT 10.1 (Noldus).

## Direct interaction test

Direct interaction was tested as described previously[80]. Mice underwent the open-field test in direct interaction box were tested. Two sex- and genotype-matched mice that never met each other were placed on the opposite edges of the custom-built box (33 × 33 × 22 cm) used for the open-field test. Time spent in nose-to-nose interaction, nose-to-body interaction, and following was measured. Sniffing on the nose or whisker of the partner mouse was counted as nose-to-nose interaction. Sniffing on the other body parts was nose-to-body interaction. Time spent in chasing after the partner mouse was counted as following. Following started within 3 s of partner mouse movement was measured. For conditional KO mice, a subject mouse was paced on the edge of the box with age-matched stranger male C3H strain mouse, followed by 10 min recordings and analysis. Social interaction initiated by the subject mice was counted.

## Three-chamber test

The three-chamber test was performed as previously described[81]. The white acrylic three-chambered apparatus (60 cm × 40 cm × 20 cm) including two small containers in the corner of the side chambers was used. Sex- and age- matched group housed (4–6 mice) mice (129S1/SvlmJ strain) were prepared to give a role as stranger. There were three different sessions with each of 10 min, habituation, stranger and object discrimination session (S1-O) and last old and new stranger discrimination session (S1-S2). During habituation session, subject mice was allowed to freely explore the chamber with empty containers. After habituation, stranger mice were located in left or right container of apparatus in random manner. Blue cylindrical object was located in the other container. During location, subject mice was located in center chamber of apparatus, and not allowed to move to, or watch other chambers. After that, subject mice could freely interact with stranger or blue cylindrical object. (S1-O). Finally, object was replaced with new stranger mouse with same way in S1-O, and subject mice could freely interact with familiar mouse (S1) and novel mice (S2). During all of these sessions, top side view of apparatus was video recorded, and sniffing time and chamber location time of objective mice in each sessions were analyzed using Ethovision XT 10.1 (Noldus).

## Pup ultrasonic vocalization

Pup ultrasonic vocalizations (USVs) were measured using an ultrasound microphone (Avisoft) placed above the testing arena. Each pup separated from its mother was located into a glass container. Pup USVs were recorded during 200 s. Avisoft SASlab Pro was used to generate spectrograms and analyze the data.

## Elevated plus-maze test

The elevated plus maze (EPM) apparatus with two closed arms (30 × 5 × 30 cm), two open arms (30 × 5 × 0.5 cm), and a neutral center zone linked with four different arms was located 50 cm above the ground. The light intensity of the center zone was set at 100 lux. Subject mice was located in the center zone, and allowed to freely explore the apparatus for 10 min. The center point of mouse body was used to determine the location of mouse in the apparatus. Time spent in and frequency of entry to open/closed arms were measured using Ethovision XT 10.1 (Noldus).

## Light-dark test

The light-dark apparatus contained two chambers (20 × 30 × 20 cm, 300 lux for light chamber, 20 × 13 × 20 cm, 0 lux for dark chamber) and a 5 cm wide entrance between the two chambers. Subject mice were located in the light chamber with their heads pointing toward the opposite wall from the dark chamber, and allowed to freely explore the whole apparatus for 10 min. Time spent in the light chamber was analyzed using Ethovision XT 10.1 (Noldus).

## Novel object recognition test

Novel object recognition memory test was done during three consecutive days. On day 1, mice were individually located in the center region of the white acryl circular chamber for 30 min habituation. On day 2, the subject mouse was allowed to explore two identical objects symmetrically localized in the chamber for 10 min. On day 3, one of the two objects was replaced by a novel object with different shape and texture in random manner, and the mouse was allowed to freely explore the two objects for 10 min. Sniffing time for novel object (N) and familiar object (F) was analyzed using Ethovision XT 10.1 (Noldus).

## Forced swim test

A glass beaker filled with 1500 ml of fresh water was used for the forced swim test. Subject mice were placed on the water, and their movement was recorded from a side camera for 6 min. Immobilization time, defined by floating or the absence of active limb or body movements, was measured by a blind manner.

## Brain lysates

Whole-brain and mPFC samples from *Arid1b*[+/-] mice and WT littermates were homogenized in ice-cold brain homogenization buffer (0.32 M sucrose, 10 mM HEPES, pH7.4, 2 mM EDTA, 2 mM EGTA, protease inhibitors, phosphatase inhibitors). Then, the homogenized brains were centrifuged at $1200 \times g$ for 10 min. The pellet was then resuspended in homogenization buffer and used as a nuclear (P1) fraction. The supernatant was additionally centrifuged at $12,000 \times g$ for 10 min. The pellet was resuspended in homogenization buffer and used as a crude synaptosomal (P2) fraction.

## Immunoblot analysis

HRP-conjugated and fluorescent secondary antibody signals were captured by Odyssesy Fc Dual Mode Imaging System (Li-COR). Immunoblot signals were quantified using Image Studio Lite (Ver 3.1). Following antibodies were purchased: Arid1b (Abcam, ab57461, 1:1000), HDAC4 (Abcam, ab111318, 1:1000), β-catenin (BD Transduction, 610154, 1:2000), pβ-catenin (Ser-552, Cell Signaling, 9566, 1:1000), pβ-Catenin (Ser-675, Cell Signaling, 4176, 1:1000), GAPDH (Cell Signaling, 2118L, 1:2000), and PSD-95 (NeuroMab, 75-028, 1:1000), SynGAP (Invitrogen, PA1-046, pan-SynGAP antibody, 1:1000), Arc (Synaptic system, 156 003, 1:500), MEF2A (Cell Signaling, 9736S, 1:500), pMEF2A (Ser-408, Cell Signaling, 9737S, 1:500), HRP conjugated anti-Rb IgG (Jackson, Cat # 711-035-152, 1:10,000), HRP conjugated anti-Ms IgG (Jackson, Cat # 715-035-150, 1:10,000), IRDye 800CW conjugated anti-Ms IgG (Li-Cor, Cat # 926-32212, 1:10000).

## Immunohistochemistry

Adult mice (P42 or P56) underwent cardiac perfusion using 1% heparin and 4% paraformaldehyde. Dissected brains were stored in 4% paraformaldehyde overnight. Coronal sections (50 μm) were prepared using a vibratome (Leica). Sections were permeabilized with 1× phosphate-buffered saline (PBS) with 0.5% Triton X-100, blocked with 5% serum (goat/donkey), and incubated with primary antibodies (NeuN, Milipore, MAB377, 1:500; Cux1, Santa Curz, sc-13024, 1:500; FoxP2, Abcam, ab16046, 1:500; parvalbumin, Millipore, MAB1572, 1:1000) and Alexa Fluor 594 AffiniPure Donkey Anti-ms IgG (H+L) (Jackson, Cat # 715-585-150, 1:1000), Alexa Fluor 594 AffiniPure Donkey Anti-rb IgG (H+L) (Jackson, Cat # 711-585-152, 1:1000), and Alexa Fluor 488 AffiniPure Donkey Anti-ms IgG (H+L) (Jackson, Cat # 715-545-151, 1:1000). For Nissl staining for gross morphological analysis, brain slices were incubated with diluted Nissl (ThermoFisher; 1:1000) for 30 min. The shapes of brain structures were marked manually, and each of area was measured in a blind manner. Images were acquired using LSM 780 (Carl Zeiss) or AxioScan.Z1 (Carl Zeiss).

## Brain 5-HT measurement

5-HT analysis was performed according to previously reported UPLC-MS/MS-based neurochemical profiling methods in mice[82]. Briefly, five volumes of distilled water containing 0.1% formic acid were added to mouse whole brain tissue ($n = 3–6$), after which tissue was homogenized by ultrasonication. Each brain homogenate was extracted by adding an equal volume of methanol containing 0.1% formic acid, followed by vortexing and centrifugation at $21,130 \times g$ at $4\,°C$ for 20 min. Cleared supernatants were analyzed by LC-MS/MS. Quantitative analyses of 5-HT were performed using an ultra-performance liquid chromatography system (ACQUITY UPLC System, Waters Corporation) coupled with a Xevo TQ-S triple quadrupole mass spectrometer (Waters Corporation).

## Radioisotope in situ hybridization

Mouse brain sections (14 μm thick) at embryonic day (E18) and postnatal days (P1, P7, P14, P21, and P56) were prepared using a cryostat (Leica CM 1950). Hybridization probe specific for mouse Arid1b mRNA was prepared using the following regions: nt 6487–6786 of Arid1b (NM_001085355.1). Antisense riboprobe was generated using 35S-uridine triphosphate (UTP) and the Riboprobe system (Promega).

## Fluorescent in situ hybridization (FISH)

In brief, frozen mouse brain sections (14 μm thick) were cut coronally through the hippocampus and dorsal raphe and thaw-mounted onto Superfrost Plus Microscope Slides (Fisher Scientific). The sections were fixed in 4% paraformaldehyde, followed by dehydration in increasing concentrations of ethanol and protease digestion. For hybridization, the sections were incubated in different amplifier solutions in a HybEZ hybridization oven (Advanced Cell Diagnostics) at $40\,°C$. The probes used in these studies were three synthetic oligonucleotides complementary to the nt sequence 1420–1864 of Mm-Arid1b-01-C1, 452–1378 of Mm-Slc6a4-C2, 62–3113 of Mm-Gad1-C2, 552–1506 of Mm-Gad2-C3, 464–1415 of Mm-Slc17a7 (Vglut1)-C2, and 1986–2998 of Mm-Slc17a6 (Vglut2)-C3 (Advanced Cell Diagnostics). The labeled probes were conjugated to Alexa Fluor 488, Altto 550, or Atto 647. The sections were hybridized with probe mixtures at $40\,°C$ for 2 h. Nonspecifically hybridized probes were removed by washing the sections in 1× wash buffer, and the slides were treated with Amplifier 1-FL for 30 min, Amplifier 2-FL for 15 min, Amplifier 3-FL for 30 min, and Amplifier 4 Alt B-FL for 15 min. Each amplifier was removed by washing with 1× wash buffer. The slides were viewed and photographed using TCS SP8 Dichroic/CS (Leica).

## Sholl analysis

Pyramidal neuron of hippocampal CA1 from adult Arid1b HT-Thy1-EGFP (P42) was imaged. Images were acquired using LSM 780 (Carl Zeiss), and 3D neuronal images were built using neuTube[83]. SWC files were imported, and neuronal branches were analyzed with Sholl analysis plugin built in ImageJ.

## RNA sequencing and analyses

Mouse brains dissected and immersed in RNAlater solution (Ambion, AM7020) to stabilize RNA. RNA extraction, library preparation, cluster generation, and sequencing were carried by Macrogen Inc. (Seoul, Korea). Briefly, poly-A-mRNAs were purified Using poly-T-oligo-attached magnetic beads. Then, the purified RNAs were fragmented using divalent cations under elevated temperature. RNA concentrations were calculated using Quant-IT RiboGreen (Invitrogen, R11490), and the integrity of total RNAs were determined using TapeStation RNA screenTape (Agilent Technologies). Only high-quality RNAs (RIN > 7.0) were used for cDNA library construction using Illumina TruSeq mRNA Sample Prep kit (Illumina). First-strand cDNA synthesis was carried out using SuperScript II reverse transcriptase (Invitrogen) and random primers, followed by second-strand cDNA synthesis using

DNA polymerase I and RNase H. The cDNA fragments underwent end-repair process, single 'A' base addition, and indexing adapter ligation. These products were purified and amplified using PCR to create the final cDNA library. The cDNA library was quantified by qPCR using qPCR Quantification Protocol Guide (KAPA Library Quantification kits for Illumina Sequencing platforms) and TapeStation D1000 ScreenTape (Agilent Technologies). Indexed libraries were submitted to an Illumina NovaSeq (Illumina), and the paired-end ($2 \times 100$ bp) sequencing was conducted by Macrogen Inc.

Transcript abundance was estimated with Salmon (v1.1.0) in Quasi-mapping-based mode onto the Mus musculus genome (GRCm38) with GC bias correction (--gcBias). Quantified gene-level abundance data was imported to R (v.3.5.3) using tximport package, and differential gene expression analysis was conducted using R/Bioconductor DEseq2 (v1.30.1). Normalized read counts were computed by dividing the raw read counts by size factors and fitted to a negative binomial distribution. The $P$ values were adjusted for multiple testing with the Benjamini–Hochberg correction. Mouse gene names were converted to human homologs using the Mouse Genome Informatics (MGI) database (http://www.informatics.jax.org/homology.shtml).

Gene set enrichment analysis was carried out using GSEAPreranked (gsea-3.0.jar) module on gene set collections downloaded from Molecular Signature Database (MsigDB) v7.0. Preranked list was generated from all gene lists, ranked by $-\log_{10}(P\ value)$ multiplied by sign of fold change. Recommended setting (1000 permutations, and a classic scoring scheme) was applied. Genesets with a false discovery rate (FDR) of less than 0.05 were considered as significantly enriched. Cytoscape App EnrichmentMap (version 3.3.1) was used to visualize gene set enrichment. Each nodes represent genesets, and their sizes are inversely proportional to FDR values. Network clustering, annotation, and layout design were performed using Cytoscape App Autoannotate (version 1.3.3.) and Cytoscape (version 3.8.0).

## Statistical analysis

Data were presented as means with standard error of mean (SEM). Statistical analyses were performed using GraphPad Prism software. Details on the mice used (species, sex, number, and age of animals) and statistical details are summarized in Supplementary Table 1.

## Reporting summary

Further information on research design is available in the Nature Research Reporting Summary linked to this article.

## Data availability

RNA-Seq results have been deposited in the Gene Expression Omnibus (GEO) database at the National Center for Biotechnology Information (NCBI) under the accession code of GSE203344 (SuperSeries) and the following sub-series codes (GSE203214 for P3 whole brain, GSE189653 for P10 whole brain, GSE203249 for P10 mPFC, GSE189653 for whole-brain P120 WT/mutant mice treated with fluoxetine and vehicle; GSE203343 for mPFC P120 WT/mutant mice treated with fluoxetine and vehicle). Raw Data for the main and supplementary figures showing quantitative results are provided in the Source Data file. Source data are provided with this paper.

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

## Acknowledgements

This work was supported by the National Research Foundation of Korea (NRF-2017R1A5A2015391 to Y.C.B.; NRF-2020M3E5D9080794 to Hyun K.), and the Institute for Basic Science (IBS-R002-D1 to E.K.).

## Author contributions

H.K. and D.K. performed behavioral experiments, and immunoblot experiments; H.K., D.K., and H. Kang performed RNA-Seq experiments; H.K., K.K., J.D.R., and Y.K. performed electrophysiological experiments; Y.C. performed electron microscopy experiments; E.Y. performed FISH experiments; H.K. and D.K. maintained Arid1b-mutatnt mice; K.K. and H.K. performed immunohistochemistry experiments; S.S.K. performed LC-MS experiments; Y.C.B., Hyun K., S.A., and E.K. designed research and wrote the manuscript.

## Competing interests

The authors declare no competing interests.
