## [Peer Review File · Nature Communications]

Early postnatal serotonin modulation prevents adult-stage deficits in *Arid1b*-deficient mice through synaptic transcriptional reprogrammingREVIEWER COMMENTS

Reviewer #1 (Remarks to the Author):

In this paper Kim et al. deeply characterize the Arid1b+/- mice and found that these mice show typical autistic-like behavior at juvenile and adult ages associated with decreases in excitatory synaptic density and transmission in layer 2/3 pyramidal neurons in the prelimbic region of the mPFC.

Interestingly they also found that chronic treatment with fluoxetine, a selective serotonin-reuptake inhibitor, during the first three postnatal weeks is sufficient to prevent synaptic and behavioral deficits in adults. These rescues were complemented by transcriptomic changes, including upregulation of FMRP targets and normalization of HDAC4/MEF2A-related transcriptional regulation of the synaptic proteins, SynGAP1 and Arc.

Thus these results indicate that chronic serotonergic modulation during critical early postnatal periods could prevent synaptic and behavioral deficits in adult Arid1b+/- mice through a specific drug dependent transcriptional reprogramming.

This is a very complete and well presented story that clearly describes the function of an ASD related gene Arid1b and the possibility to rescue the altered synaptic deficit by early treatment with fluoxetine, a selective serotonin-reuptake inhibitor.

Therefore this manuscript can be recommended for publication after the authors will address the following concerns.

1) To further confirm that only early postnatal serotonin modulation prevents adult-stage deficits in Arid1b-deficient mice it will be interesting to evaluate if fluoxetine treatment in adults is able or unable to rescue behavioral and/or electrophysiological defects of the Arid1b+/- mice. Even if this is not a critical experiment for sustaining the main conclusions of the paper, it might help to better underline the importance of early treatment for obtaining long lasting effects in adults.

2) A very complete analysis of morphological and electrophysiological analysis was performed in the prefrontal cortex showing also specific layer alterations in the synaptic electrophysiological properties. How can be explained these selective layer specific synaptic alterations? Is this specific layer alteration potentially present in other cortex regions?

3) In a previous publication describing an Arid1b-deficient mouse, it was showed the deletion of Arid1b causes a specific reduction of the number of inhibitory neurons. In this novel Arid1b-deficient mouse it seems that the inhibitory synapses are not affected. For this reason, it will be important to count the number of inhibitory neurons in these mice and to confirm that the number of inhibitory synapses in the layer 2/3 pyramidal neurons is not reduced.

Reviewer #2 (Remarks to the Author):

The manuscript by Kim et al reports that chronic treatment of Arid1b+/- mice with fluoxetine during the first three postnatal weeks prevents synaptic and behavioral deficits in adults, which is accompanied by transcriptomic changes, including upregulation of FMRP targets and normalization of HDAC4/MEF2A-related transcriptional regulation of the synaptic proteins, SynGAP1 and Arc. This comprehensive study has provided a potential treatment avenue for Arid1b haploinsufficiency-induced phenotypes, which is important to the autism field. There are a few concerns that need to be addressed.

1. It is surprising that Arid1b+/- mice (P10) show a strong transcriptomic pattern that

have the opposite direction of ASD ("reverse-ASD") (Fig. 2, upregulation of synapse-related genes; downregulation of microglia/mitochondria-related genes), which is opposite to most ASD models. Such transcriptional changes at an early stage are also opposite to what is expected to underlie the later synaptic and behavioral phenotypes in *Arid1b*^{+/-} mice (Fig. 3, Fig. 1). These transcriptomic data were from whole-brain of P10 mice, however, different brain regions and cell types at different time points could have distinct transcriptional changes. It is important to verify the mRNA changes of key ASD-related genes in specific brain regions at the early stage and the ages when synaptic and behavioral phenotypes were measured.

2. Fig. 3 show the reduction of PFC layer 2/3 mEPSC frequency and excitatory synapse density in *Arid1b*^{+/-} mice. What is causing this synaptic loss, which is opposite to the "upregulation of synapse-related genes" (Fig. 2)?

3. Considering the prior finding that *Arid1b* haploinsufficiency disrupts cortical interneuron development (Ref 16), it is important to examine more interneuron-related changes other than mIPSC, such as interneuron markers, spontaneous IPSC, or evoked IPSC.

4. Fig. 4 shows the effect of early fluoxetine treatment (P3-21) on behavioral deficits. Does the treatment have to be so early? Does a later treatment (e.g. P21-P40) have any effect?

5. The effect of fluoxetine on social interaction in *Arid1b*^{+/-} mice (Fig. 4b) is very mild (n=6, p=0.0455), more mice need to be tested to improve the statistical power and rigor. Does early fluoxetine treatment have any effect on social communication?

6. Fig. 5, what is causing the increase of mEPSC frequency and excitatory synapse density in fluoxetine-treated *Arid1b*^{+/-} mice? Among the upregulated synapse-related genes by fluoxetine (Fig. 6), what might explain the finding of Fig. 5?

7. Fig. 7c, e, the changes of HDAC4, SYNGAP, ARC in whole brains of *Arid1b*^{+/-} mice without or with fluoxetine treatment look very mild from the blots. It is better to use nuclear or synaptic fraction of a specific brain region (e.g. PFC) for the detection of these proteins. The mRNA levels of these genes should be measured for transcriptional reprogramming by fluoxetine treatment.

8. Fig. 8 is not showing the involvement of serotonin in the effects of fluoxetine treatment, contradictory to the conclusion that it is "serotonergic modulation during critical early postnatal periods" that prevents adult-stage deficits in *Arid1b*^{+/-} mice.

9. It is confusing whether fluoxetine treatment promotes an ASD-like or a reverse-ASD transcriptomic pattern. While the focus is placed on fluoxetine-induced upregulation of FMRP target gene set and the downregulation of SynGAP1 and Arc (reverse-ASD), it is unclear why the fluoxetine-induced upregulation of synapse-related genes (ASD-like) is not considered.

Reviewer #3 (Remarks to the Author):

In this manuscript, Kim et al. used *Arid1b* haploinsufficient mice to investigate the mechanisms underlying phenotypes of *Arid1b*-deficient mice at an early stage. Here, the authors demonstrate that *Arid1b*-deficient mice display autistic-like behavior that is accompanied by reduction in excitatory synaptic density and transmission, all of which were rescued by chronic fluoxetine treatment. Overall, the manuscript is well structured, and the methodology appears sound. However, the information provided in

both the introduction and the discussion needs to be strongly expanded. Additionally, this manuscript raises some major questions that should be addressed before being acceptable for publication. These points are detailed below:

- 1- Given that *Arid1b* is considered a high confidence ASD-risk gene, it would be beneficial to the reader to introduce autism in the introduction for example by defining ASD and the challenges associated with ASD gene discovery and how ASD-risk genes were discovered etc.
- 2- What is the association of *Arid1b* and ASD? What are the known ASD-risk mutations within *Arid1b* locus? What are the impacts those mutations on the functionality.
- 3- Authors need a clear rationale as to why they decided to study *Arid1b* in the first place.
- 4- RNAseq experiments were performed on whole brain. Since ASD loss of function mutations are enriched in medial prefrontal cortex, authors must investigate the transcriptomic changes associated with *Arid1b* haploinsufficiency in mPFC as it relates to disease risk.
- 5- A volcano plot for the differentially expressed genes must be provided to illustrate the distribution of the differentially expressed genes by using $FDR < 0.05$ as the cutoff for the statistical significance.
- 6- Autism manifests itself by three years of age in humans, the extrapolation would be P3 neonatal stages in mice. However, transcriptomic analyses were performed on P10 mice. Authors need to address this in the text. Also, they need to generate RNAseq data from neonatal ages to investigate how early the HDAC4/MEF2A pathways are affected.
- 7- The findings from EM and miniature recordings are interesting, however, is this phenotype persistent into adulthood? Authors investigated the changes at P42, which is late adolescence, young adult animals. These studies must be conducted in much older animal P120 or older to demonstrate whether the phenotype persists and whether it can be rescued.
- 8- The reduction in excitatory miniature recordings are from layers 2/3 with no change in layer 5 neurons. However, there is no mention of layer 6 neurons! What happens to the firing properties of the layer 6 neurons in the *Arid1b* deficient?
- 9- In line 183, the authors mention "*Arid1b* mRNA was detected in both glutamatergic and GABAergic neurons in cortical areas". This is a very vague statement as "cortical areas" does not mean anything. Authors should be explicit on the neuroanatomy to improve the interpretation of the data.
- 10- The rescue experiments with chronic fluoxetine treatment are very interesting. Though, the authors need to investigate whether these phenotypes could be rescued in the adult animals with either chronic or acute (intraperitoneal injections) treatments.
- 11- Authors need to explicitly state how they controlled the dosing?
- 12- The authors performed RNAseq on the fluoxetine and vehicle treated animals, though they used whole-brain samples. As mentioned previously, in order to have relevance to ASD-risk, the authors must focus their transcriptomic search to medial prefrontal cortex (mPFC).
- 13- The pathways that are mentioned here are speculative. What is fluoxetine mode of action and how does it impact/regulate HDAC4/MEF2A pathway.

Point-by-point responses to reviewers' comments

Reviewer #1 (Remarks to the Author):

In this paper Kim et al. deeply characterize the *Arid1b*^{+/-} mice and found that these mice show typical autistic-like behavior at juvenile and adult ages associated with decreases in excitatory synaptic density and transmission in layer 2/3 pyramidal neurons in the prelimbic region of the mPFC.

Interestingly they also found that chronic treatment with fluoxetine, a selective serotonin-reuptake inhibitor, during the first three postnatal weeks is sufficient to prevent synaptic and behavioral deficits in adults. These rescues were complemented by transcriptomic changes, including upregulation of FMRP targets and normalization of HDAC4/MEF2A-related transcriptional regulation of the synaptic proteins, SynGAP1 and Arc.

Thus these results indicate that chronic serotonergic modulation during critical early postnatal periods could prevent synaptic and behavioral deficits in adult *Arid1b*^{+/-} mice through a specific drug dependent transcriptional reprogramming.

This is a very complete and well presented story that clearly describes the function of an ASD related gene *Arid1b* and the possibility to rescue the altered synaptic deficit by early treatment with fluoxetine, a selective serotonin-reuptake inhibitor.

Therefore this manuscript can be recommended for publication after the authors will address the following concerns.

→ We appreciate the encouraging summary of the reviewer and tried to fully address the comments, which were very helpful in improving the manuscript.

1) To further confirm that only early postnatal serotonin modulation prevents adult-stage deficits in *Arid1b*-deficient mice it will be interesting to evaluate if fluoxetine treatment in adults is able or unable to rescue behavioral and/or electrophysiological defects of the *Arid1b*^{+/-} mice. Even if this is not a critical experiment for sustaining the main conclusions of the paper, it might help to better underline the importance of early treatment for obtaining long lasting effects in adults.

→ In response, we attempted adult-stage (P100–200; not early-stage [P21–40]) chronic treatment of fluoxetine and found that it does not rescue social interaction or self-grooming, although it moderately decreased locomotor activity (**Supplementary Fig. 11**), similar to the moderately decreased locomotion by early chronic fluoxetine treatment (**Supplementary Fig. 10**).

2) A very complete analysis of morphological and electrophysiological analysis was performed in the prefrontal cortex showing also specific layer alterations in the synaptic electrophysiological properties. How can be explained these selective layer specific synaptic alterations? Is this specific layer alteration potentially present in other cortex regions?

→ In response, we tested layer 2/3 and layer 5 pyramidal neurons in the anterior

cingulate cortex/ACC (in addition to mPFC) and found that mEPSCs were not altered in either layer 2/3 or layer 5 (**Supplementary Fig. 8d,e**), suggesting the decreased mEPSC frequency in layer 2/3 pyramidal neurons are specific for the mPFC.

3) In a previous publication describing an *Arid1b*-deficient mouse, it was showed the deletion of *Arid1b* causes a specific reduction of the number of inhibitory neurons. In this novel *Arid1b*-deficient mouse it seems that the inhibitory synapses are not affected. For this reason, it will be important to count the number of inhibitory neurons in these mice and to confirm that the number of inhibitory synapses in the layer 2/3 pyramidal neurons is not reduced.

→ In response, we performed the following two experiments. First, we measured the density of parvalbumin-positive GABAergic neurons in the mPFC (prelimbic area) and found no genotype difference between WT and *Arid1b*^{+/-} mice (**Supplementary Fig. 8h**).

Second, we performed EM analysis to measure the density of GABAergic synapses and found that there is no genotype difference between WT and *Arid1b*^{+/-} mice (**Supplementary Fig. 8g**).

Reviewer #2 (Remarks to the Author):

The manuscript by Kim et al reports that chronic treatment of *Arid1b*^{+/-} mice with fluoxetine during the first three postnatal weeks prevents synaptic and behavioral deficits in adults, which is accompanied by transcriptomic changes, including upregulation of FMRP targets and normalization of HDAC4/MEF2A-related transcriptional regulation of the synaptic proteins, SynGAP1 and Arc. This comprehensive study has provided a potential treatment avenue for *Arid1b* haploinsufficiency-induced phenotypes, which is important to the autism field. There are a few concerns that need to be addressed.

→ We appreciate the encouraging summary of the reviewer and tried to fully address the comments, which were very helpful in improving the manuscript.

1. It is surprising that *Arid1b*^{+/-} mice (P10) show a strong transcriptomic pattern that have the opposite direction of ASD (“reverse-ASD”) (Fig. 2, upregulation of synapse-related genes; downregulation of microglia/mitochondria-related genes), which is opposite to most ASD models. Such transcriptional changes at an early stage are also opposite to what is expected to underlie the later synaptic and behavioral phenotypes in *Arid1b*^{+/-} mice (Fig. 3, Fig. 1). These transcriptomic data were from whole-brain of P10 mice, however, different brain regions and cell types at different time points could have distinct transcriptional changes. It is important to verify the mRNA changes of key ASD-related genes in specific brain regions at the early stage and the ages when synaptic and behavioral phenotypes were measured.

→ In response, we performed two following additional RNA-Seq analyses. First, we performed RNA-Seq analysis using mPFC samples from *Arid1b*^{+/-} mice at P10 and found strong ASD-like transcriptomic patterns (i.e., decreases in synapse-, neuron-,

and ASD-related gene expressions, and increased astrocyte/microglial gene expression) (**Fig. 2**; both mPFC and whole-brain results are shown). This suggests that mPFC and whole-brain transcriptomic patterns are different, as correctly predicted by the reviewer.

Second, we performed RNA-Seq analysis of mPFC samples from fluoxetine-treated *Arid1b*^{+/-} mice at P120 and found patterns that are partly distinct from the whole-brain pattern (i.e. increased synaptic gene expression) but also those that are similar to the whole-brain pattern (i.e., increased FMRP target gene expression) (**Supplementary Fig. 12**). This suggests that FMRP targets may be an important compensatory mechanism that are shared in the whole-brain and mPFC.

2. Fig. 3 show the reduction of PFC layer 2/3 mEPSC frequency and excitatory synapse density in *Arid1b*^{+/-} mice. What is causing this synaptic loss, which is opposite to the “upregulation of synapse-related genes” (Fig. 2)?

→ We think that the decreased synaptic gene expression observed in the mPFC (not the whole brain) is responsible for the decrease in mEPSC frequency in mPFC layer 2/3 pyramidal neurons. We mentioned this in the revised Results.

3. Considering the prior finding that *Arid1b* haploinsufficiency disrupts cortical interneuron development (Ref 16), it is important to examine more interneuron-related changes other than mIPSC, such as interneuron markers, spontaneous IPSC, or evoked IPSC.

→ In response, we performed the following three experiments. First, we measured the density of parvalbumin-positive GABAergic neurons in the mPFC (prelimbic area) and found no genotype difference between WT and *Arid1b*^{+/-} mice (**Supplementary Fig. 8h**).

Second, we performed EM analysis to measure the density of GABAergic synapses and found that there is no genotype difference between WT and *Arid1b*^{+/-} mice (**Supplementary Fig. 8g**).

Third, we measured sIPSCs in layer 2/3 pyramidal neurons and found no genotype difference between WT and *Arid1b*^{+/-} mice (**Supplementary Fig. 8f**).

4. Fig. 4 shows the effect of early fluoxetine treatment (P3-21) on behavioral deficits. Does the treatment have to be so early? Does a later treatment (e.g. P21-P40) have any effect?

→ In response, we attempted adult-stage (P100–200; not early-stage [P21–40]) chronic treatment of fluoxetine and found that it does not rescue social interaction or self-grooming, although it moderately decreased locomotor activity (**Supplementary Fig. 11**), similar to the moderately decreased locomotion by early chronic fluoxetine treatment (**Supplementary Fig. 10**).

5. The effect of fluoxetine on social interaction in *Arid1b*^{+/-} mice (Fig. 4b) is very mild ($n=6$, $p=0.0455$), more mice need to be tested to improve the statistical power and rigor. Does early fluoxetine treatment have any effect on social communication?

→ We increased mouse numbers for all four groups of this experiment and found that early fluoxetine experiment rescues social interaction without affecting WT mice, as supported by two-way ANOVA statistics (**Fig. 4a,b**). We did not test if fluoxetine treatment rescues the reduced USVs observed in the mutant pups separated from their mothers because the duration of drug treatment (P3–21) lasts longer than the period during which USVs were measured (P3–11) (**Fig. 1f**).

6. Fig. 5, what is causing the increase of mEPSC frequency and excitatory synapse density in fluoxetine-treated *Arid1b*^{+/-} mice? Among the upregulated synapse-related genes by fluoxetine (Fig. 6), what might explain the finding of Fig. 5?

→ We appreciate this thoughtful question. Our data indicate that early fluoxetine increases the expression of FMRP target genes in both the whole brain and mPFC of *Arid1b*^{+/-} mice (**Fig. 6c** and **Supplementary Fig. 12c**). In addition, early fluoxetine increases the levels of the HDAC4 protein, a key FMRP target, in the whole brain and mPFC of *Arid1b*^{+/-} mice (**Fig. 7c** and **Supplementary Fig. 13a**). This promotes the nuclear localization of HDAC4 (**Fig. 7c**) and HDAC4-dependent inhibition of the MEF2A transcription factor (**Fig. 7d**), known to inhibit excitatory synapses by promoting the expression of SynGAP1 (inhibitor of Ras small GTPases) and Arc (promoter of metabotropic glutamate receptor/mGluR-mediated long-term depression). Indeed, SynGAP1 levels are decreased in early fluoxetine-treated whole brain and mPFC of *Arid1b*^{+/-} mice (**Fig. 7e** and **Supplementary Fig. 13b**). However, we have to point out that many other FMRP targets and downstream mediators of HDAC4/MEF2A would act cooperatively to rescue the behavioral phenotypes. We further clarified these points in the revised Discussion.

7. Fig. 7c, e, the changes of HDAC4, SYNGAP, ARC in whole brains of *Arid1b*^{+/-} mice without or with fluoxetine treatment look very mild from the blots. It is better to use nuclear or synaptic fraction of a specific brain region (e.g. PFC) for the detection of these proteins. The mRNA levels of these genes should be measured for transcriptional reprogramming by fluoxetine treatment.

→ We appreciate this careful comment. In response, we repeated these immunoblot experiments using mPFC and nuclear/synaptic samples and found similar changes (decreased HDAC4 levels in the nuclear/P1 fraction and increased SynGAP levels in the P2/crude synaptosomal fraction), although Arc levels were not changed in the mutant samples (**Supplementary Fig. 13a-c**).

There were no significant changes in the levels of HDAC4, SynGAP1, and Arc mRNAs in the whole brain and mPFC (**Supplementary Fig. 13d**), although the directions of the changes tended to be similar to those observed in the proteins, i.e., HDAC4.

8. Fig. 8 is not showing the involvement of serotonin in the effects of fluoxetine treatment, contradictory to the conclusion that it is “serotonergic modulation during critical early postnatal periods” that prevents adult-stage deficits in *Arid1b*^{+/-} mice.

→ We appreciate this careful comment. It is indeed the early fluoxetine treatment or serotonergic receptor modulation, but not serotonergic modulation, that rescued the

phenotypes. Accordingly, we changed related texts throughout the manuscript.

9. It is confusing whether fluoxetine treatment promotes an ASD-like or a reverse-ASD transcriptomic pattern. While the focus is placed on fluoxetine-induced upregulation of FMRP target gene set and the downregulation of SynGAP1 and Arc (reverse-ASD), it is unclear why the fluoxetine-induced upregulation of synapse-related genes (ASD-like) is not considered.

→ We appreciate this thoughtful comment. The comparison of the RNA-Seq results from the whole-brain and mPFC of *Arid1b*^{+/-} mice early treated with fluoxetine indicate a common increase in the expression of FMRP target genes, but an increase in synapse-related genes only in the whole brain but not mPFC (**Fig. 6; Supplementary Fig. 12**). This was clarified in the revised Results.

Reviewer #3 (Remarks to the Author):

In this manuscript, Kim et al. used *Arid1b* haploinsufficient mice to investigate the mechanisms underlying phenotypes of *Arid1b*-deficient mice at an early stage. Here, the authors demonstrate that *Arid1b*-deficient mice display autistic-like behavior that is accompanied by reduction in excitatory synaptic density and transmission, all of which were rescued by chronic fluoxetine treatment. Overall, the manuscript is well structured, and the methodology appears sound. However, the information provided in both the introduction and the discussion needs to be strongly expanded. Additionally, this manuscript raises some major questions that should be addressed before being acceptable for publication. These points are detailed below:

→ We appreciate the encouraging summary of the reviewer and tried to fully address the comments, which were very helpful in improving the manuscript.

1- Given that *Arid1b* is considered a high confidence ASD-risk gene, it would be beneficial to the reader to introduce autism in the introduction for example by defining ASD and the challenges associated with ASD gene discovery and how ASD-risk genes were discovered etc.

→ We appreciate this comment and tried to modify the revised introduction section as suggested.

2- What is the association of *Arid1b* and ASD? What are the known ASD-risk mutations within *Arid1b* locus? What are the impacts those mutations on the functionality.

→ We appreciate this comment again and tried to mention these aspects in the revised introduction section.

3- Authors need a clear rationale as to why they decided to study *Arid1b* in the first place.

→ We appreciate this comment again. ARID1B represents one of the most frequently mutated genes in ID, ASD, and Coffin-Siris Syndrome. We tried to clarify this in the revised introduction section.

4- RNAseq experiments were performed on whole brain. Since ASD loss of function mutations are enriched in medial prefrontal cortex, authors must investigate the transcriptomic changes associated with *Arid1b* haploinsufficiency in mPFC as it relates to disease risk.

→ In response, we performed two following additional RNA-Seq analyses. First, we performed RNA-Seq analysis using PFC samples from *Arid1b*^{+/-} mice at P10 and found strong ASD-like transcriptomic patterns (i.e., decreased synaptic gene expression and decreased ASD-risk gene expression, decreased neuronal gene expression, and increased astrocyte/microglial gene expression) (**Fig. 2**; both PFC and whole-brain results are shown). This suggests that PFC and whole-brain transcriptomic patterns are different.

Second, we performed RNA-Seq analysis of PFC sample from *Arid1b*^{+/-} mice at P120 and found patterns that are different from the whole-brain pattern (i.e. increased synaptic gene expression) but also those that are similar to the whole-brain pattern (i.e., increased FMRP target gene expression) (**Supplementary Fig. 12**). This suggests that FMRP targets may be an important compensatory mechanism that are shared in the whole-brain and PFC.

5- A volcano plot for the differentially expressed genes must be provided to illustrate the distribution of the differentially expressed genes by using FDR<0.05 as the cutoff for the statistical significance.

→ In response, we now show volcano plots of all DEGs (FDR <0.05) from various RNA-Seq analyses (P3 whole brain, P10 whole brain and PFC, and P120 whole brain and PFC of fluoxetine-treated/untreated mice) (**Supplementary Fig. 6**). We did not attempt additional analyses of the DEG such as DAVID analysis for the small numbers of DEGs (indicated above the volcano plots).

6- Autism manifests itself by three years of age in humans, the extrapolation would be P3 neonatal stages in mice. However, transcriptomic analyses were performed on P10 mice. Authors need to address this in the text. Also, they need to generate RNAseq data from neonatal ages to investigate how early the HDAC4/MEF2A pathways are affected.

→ In response, we performed RNA-Seq analyses using *Arid1b*^{+/-} whole brains at P3 (**Supplementary Fig. 7**). The overall patterns of the P3 transcriptome in the *Arid1b*^{+/-} whole brain were similar to those observed in the P10 whole-brain transcriptome (**Fig. 2**).

7- The findings from EM and miniature recordings are interesting, however, is this phenotype persistent into adulthood? Authors investigated the changes at P42, which is late adolescence, young adult animals. These studies must be conducted in

much older animal P120 or older to demonstrate whether the phenotype persists and whether it can be rescued.

→ We agree with the reviewer's point that it is important to see if the synaptic deficits (mEPSC and EM results) observed at P42 are also observed in older animals. Although we do not have such measurements in naïve old mice, we believe that the vehicle-treated old Arid1b^{+/-} mice at ~P100–120 are similar to naïve old mice. Importantly, these mice show decreases in mEPSC frequency and dendritic spine density (EM) (**Fig. 5**).

8- The reduction in excitatory miniature recordings are from layers 2/3 with no change in layer 5 neurons. However, there is no mention of layer 6 neurons! What happens to the firing properties of the layer 6 neurons in the Arid1b deficient?

→ In response, we measured mEPSCs from layer 6 pyramidal neurons and found a no genotypes difference in mEPSC frequency or amplitudes between WT and Arid1b^{+/-} mice at P56 (**Supplementary Fig. 8c**).

9- In line 183, the authors mention "Arid1b mRNA was detected in both glutamatergic and GABAergic neurons in cortical areas". This is a very vague statement as "cortical areas" does not mean anything. Authors should be explicit on the neuroanatomy to improve the interpretation of the data.

→ We appreciate this comment and changed the description from "...in cortical areas" to "...in cortical areas, including the retrosplenial cortex (see figure legends for further details)".

10- The rescue experiments with chronic fluoxetine treatment are very interesting. Though, the authors need to investigate whether these phenotypes could be rescued in the adult animals with either chronic or acute (intraperitoneal injections) treatments.

→ In response, we attempted adult-stage (P100–200; not early-stage [P21–40]) chronic treatment of fluoxetine and found that it does not rescue social interaction or self-grooming, although it moderately decreased locomotor activity (**Supplementary Fig. 11**), similar to the moderately decreased locomotion by early chronic fluoxetine treatment (**Supplementary Fig. 10**).

11- Authors need to explicitly state how they controlled the dosing?

→ We added the following statements in Methods: "We did not directly measure the levels of fluoxetine and its metabolites in the sera of newborn mice, but a previous study employing the same drug treatment strategy has demonstrated that fluoxetine can reach the blood of newborn mice via mother's milk and reported detailed concentrations of serum fluoxetine and its metabolite during and after the treatment."

12- The authors performed RNAseq on the fluoxetine and vehicle treated animals, though they used whole-brain samples. As mentioned previously, in order to have

relevance to ASD-risk, the authors must focus their transcriptomic search to medial prefrontal cortex (mPFC).

→ In response, we performed RNA-Seq analysis of mPFC samples from fluoxetine/vehicle-treated WT and *Arid1b*^{+/-} mice at P120 and found patterns that are different from the whole-brain pattern (i.e. increased synaptic gene expression) but also those that are similar to the whole-brain pattern (i.e., increased FMRP target gene expression) (**Supplementary Fig. 12**). This suggests that FMRP targets may be an important compensatory mechanism that is shared in the whole-brain and PFC.

13- The pathways that are mentioned here are speculative. What is fluoxetine mode of action and how does it impact/regulate HDAC4/MEF2A pathway.

→ We appreciate this thoughtful comment and added the following sentence to Discussion: “However, how early fluoxetine treatment elevates HDAC4 levels remains unclear, although chromatin-related gene expressions are strongly decreased in fluoxetine-treated *Arid1b*^{+/-} mice (**Fig. 6b**), suggesting that chromatin remodeling-related transcriptional changes may be involved.”.

REVIEWERS' COMMENTS

Reviewer #1 (Remarks to the Author):

The authors fully addressed my comments.

Reviewer #2 (Remarks to the Author):

The authors have added new data and discussions to address my prior concerns. The revised manuscript is significantly improved.

Reviewer #3 (Remarks to the Author):

I have read the revised version of the manuscript thoroughly, and the detailed 8-page response to reviewers. I am very satisfied with responses to my prior critiques and definitely approve the manuscript for publication. These were addressed by provision of more data in the figures, editing of the text, particularly by providing more methodological detail. I believe that this manuscript contributes substantially novel information relevant to mechanisms mediated by Arid1b mutations. The Arid1b mutation represents an important, recurrent mutation found in ASD. Overall, I am more than satisfied with this revised manuscript. The experiments are exhaustive and the analysis and interpretation are robust, rigorous and detailed for each section.

Point-by-point response to the reviewers' comments

REVIEWERS' COMMENTS

Reviewer #1 (Remarks to the Author):

The authors fully addressed my comments.

→ We appreciate this kind comment of the reviewer.

Reviewer #2 (Remarks to the Author):

The authors have added new data and discussions to address my prior concerns. The revised manuscript is significantly improved.

→ We appreciate this kind comment of the reviewer.

Reviewer #3 (Remarks to the Author):

I have read the revised version of the manuscript thoroughly, and the detailed 8-page response to reviewers. I am very satisfied with responses to my prior critiques and definitely approve the manuscript for publication. These were addressed by provision of more data in the figures, editing of the text, particularly by providing more methodological detail. I believe that this manuscript contributes substantially novel information relevant to mechanisms mediated by Arid1b mutations. The Arid1b mutation represents an important, recurrent mutation found in ASD. Overall, I am more than satisfied with this revised manuscript. The experiments are exhaustive and the analysis and interpretation are robust, rigorous and detailed for each section.

→ We appreciate this kind comment of the reviewer and fully agree with the comment that Arid1b mutations and related mechanisms are very important in the field of ASD.